# Sortformer: A Novel Approach for Permutation-Resolved Speaker Supervision in Speech-to-Text Systems

**Taejin Park** [* 1]  **Ivan Medennikov** [* 1]  **Kunal Dhawan** [* 1]  **Weiqing Wang** [* 1]  **He Huang** [1]  **Nithin Rao Koluguri** [1]
**Krishna C. Puvvada** [1]  **Jagadeesh Balam** [1]  **Boris Ginsburg** [1]

## Abstract

*Sortformer* is an encoder-based speaker diarization model designed for supervising speaker tagging in speech-to-text models. Instead of relying solely on permutation invariant loss (PIL), Sortformer introduces Sort Loss to resolve the permutation problem, either independently or in tandem with PIL. In addition, we propose a streamlined multi-speaker speech-to-text architecture that leverages Sortformer for speaker supervision, embedding speaker labels into the encoder using sinusoidal kernel functions. This design addresses the speaker permutation problem through sorted objectives, effectively bridging timestamps and tokens to supervise speaker labels in the output transcriptions. Experiments demonstrate that Sort Loss can boost speaker diarization performance, and incorporating the speaker supervision from Sortformer improves multi-speaker transcription accuracy. We anticipate that the proposed Sortformer and multi-speaker architecture will enable the seamless integration of speaker tagging capabilities into foundational speech-to-text systems and multimodal large language models (LLMs), offering an easily adoptable and user-friendly mechanism to enhance their versatility and performance in speaker-aware tasks. The code and trained models are made publicly available through the NVIDIA NeMo Framework.

## 1. Introduction

With recent advances in deep neural networks and large language models (LLMs), automatic speech recognition (ASR) is being deployed across a broader range of industrial applications, enabling numerous new use cases. In tran-

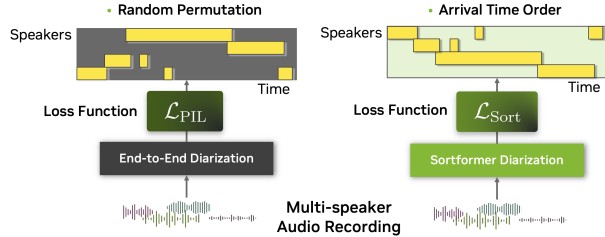

*Figure 1.* Sortformer resolves permutation problem by following the arrival-time order of the speech segments.

scription services, a growing number of applications require speaker annotations because natural language understanding (NLU) modules need to recognize speakers to gain a deeper understanding of conversations and interactions. Moreover, as modern machine learning models demand large amounts of training data, the need for automatic annotation systems has grown significantly.

The rising demand for speaker annotations underscores the need for robust speaker tagging—also known as speaker diarization, which is the process of estimating generic speaker labels by assigning audio segments to individual speakers. In the context of automatic speech recognition (ASR), multi-speaker ASR (also referred to as speaker-attributed ASR or multi-talker ASR in the literature) requires the speaker diarization process, either directly or indirectly, to transcribe spoken words with speaker annotations alongside the generated text. As ASR models continue to be streamlined and become more accurate, speaker diarization is progressively integrated into the ASR framework or performed simultaneously during the ASR decoding process, enabling rich transcription with conversational context.

Despite recent advances in speaker diarization and multi-speaker ASR, these systems have been typically trained, deployed, and evaluated separately from ASR models due to challenges such as data scarcity and application diversity. Collecting annotated multi-talker conversational speech is significantly more difficult than acquiring images or single-speaker speech data, particularly for low-resource languages or privacy-sensitive domains such as medical applications. Additionally, multi-speaker ASR use cases often require models to perform inference on multi-hour audio samples,

---

[*]Equal contribution  [1]NVIDIA, Santa Clara, USA. Correspondence to: Taejin Park <taejinp@nvidia.com>.

*Proceedings of the 42$^{nd}$ International Conference on Machine Learning*, Vancouver, Canada. PMLR 267, 2025. Copyright 2025 by the author(s).

while acquiring such long-form training data is even more challenging.

Although these cascaded multi-speaker ASR systems achieve competitive performance, optimizing or fine-tuning high-performance multi-speaker ASR systems for specific domains remains a considerable challenge, as demonstrated by evaluations such as CHiME challenges (Barker et al., 2017; 2018; Watanabe et al., 2020; Cornell et al., 2023). On the other hand, end-to-end multi-speaker ASR models without explicit speaker diarization modules have been proposed (Kanda et al., 2020b; Shi et al., 2024) which are based on the serialized output training (SOT) technique. However, training such end-to-end multi-speaker ASR systems requires speaker-annotated multi-speaker data, which is relatively scarce and challenging to collect and annotate. As a result, the performance of end-to-end multi-speaker ASR systems tends to lag behind that of cascaded systems (Kanda et al., 2022b).

To address these challenges, we propose *Sortformer*[1], introducing *Sort Loss* and techniques for bridging timestamps with text tokens. Despite the popularity of end-to-end speaker diarization systems, such speaker diarization models have not been able to be seamlessly integrated into ASR models or multimodal LLMs. To overcome this, we introduce an arrival time sorting (ATS) approach, where speaker tokens from ASR outputs and speaker timestamps from diarization outputs are sorted by arrival times to resolve permutations (see Figure 1).

Our proposed method enables multi-speaker ASR systems or multimodal LLMs to be trained or fine-tuned while significantly improving in speaker tagging accuracy with a relatively small amount of fine-tuning. A key advantage is that multi-speaker ASR training can leverage a standard token-level cross-entropy loss, facilitated by the permutation-resolved speaker supervision of the Sortformer model. This approach makes multi-speaker ASR training functionally equivalent to standard mono-speaker ASR training and fine-tuning, requiring only minimal architectural adjustments. Additionally, our method eliminates the need for word-level or segment-level timestamps, significantly reducing annotation requirements. Furthermore, Sortformer can function independently as an end-to-end speaker diarization model.

## 2. Related Works

### 2.1. Speaker Diarization

Before multi-speaker ASR gained prominence, speaker diarization handled the task of identifying *"who spoke when"* without transcription. Early systems, such as the

---

[1]https://huggingface.co/nvidia/diar_sortformer_4spk-v1

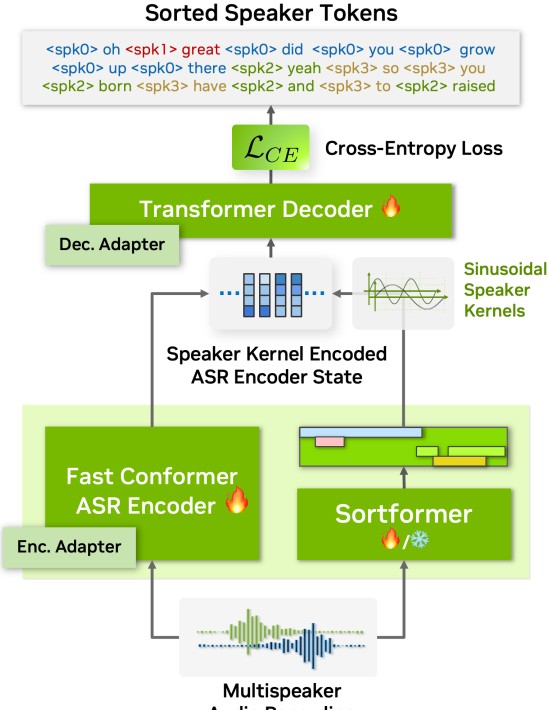

*Figure 2.* The overall dataflow of speaker supervision from Sortformer model integrated into the proposed MS-ASR system.

RT03 evaluation (Tranter et al., 2003), combined ASR word timestamps with speaker segmentations or attempted to use phrase dictionaries (Canseco-Rodriguez et al., 2004), which showed limited success. Recent advances, like TS-VAD (Medennikov et al., 2020a), revolutionized cascaded multi-speaker ASR by employing target-speaker voice activity detection into the multi-speaker ASR pipeline, influencing subsequent systems (Wang & Li, 2022; Yang et al., 2024). These approaches achieved strong results in real-world challenges (Wang et al., 2021; Niu et al., 2024), cementing their relevance in modular systems.

To streamline the speaker diarization system, a slew of end-to-end neural diarization (EEND) models were proposed, framing speaker labeling as a frame-wise classification task with permutation invariant training (PIT) loss (Yu et al., 2017b; Fujita et al., 2019). Subsequent works (Fujita et al., 2020; Takashima et al., 2021) introduced flexible output dimensions to accommodate varying numbers of speakers, while hybrid approaches like EEND-EDA (Horiguchi et al., 2020) and attention-based models (Chen et al., 2024) achieved greater accuracy.

### 2.2. Multi-speaker ASR

Early studies on multi-speaker ASR—exemplified by (Yu et al., 2017a; Qian et al., 2018)—used separate components for source separation and transcription. Jointly trainable systems (Shafey et al., 2019) later introduced speaker at-

tribution alongside text generation. Key advancements like SOT (Kanda et al., 2020b) enabled simpler models by leveraging attention mechanisms, extending to token-level SOT (t-SOT) for streaming (Kanda et al., 2022a). Recent systems, such as (Shi et al., 2024), adopt non-PIT loss schemes like dominance ranking. Strictly end-to-end systems often face limitations in handling speaker counting and domain-specific datasets (Shafey et al., 2019; Wang et al., 2024). Cascaded systems like Transcribe-to-Diarize (Kanda et al., 2022b) combine diarization and ASR with SOT, while modular systems incorporating clustering steps (Cornell et al., 2023) still show strong performance. However, such systems require extensive tuning, highlighting the need for more adaptable architectures.

## 2.3. Limitations of Previous Approaches

Despite the abundance of high-performing end-to-end diarization and ASR models (Fujita et al., 2019; Horiguchi et al., 2022a; Chen et al., 2024), there have been limited efforts to create a synergistic effect by integrating both models within a differentiable computational graph. To the best of our knowledge, our proposed system is the first to integrate an end-to-end diarization system with an end-to-end multi-speaker ASR model at the computational graph level. Challenge-winning cascaded or modular systems (Cornell et al., 2023; Medennikov et al., 2020b; Niu et al., 2024) demonstrate remarkable performance, where speaker diarization and ASR are processed sequentially with additional source separation modules (Boeddecker et al., 2018; Žmolíková et al., 2019). However, these systems are difficult to optimize because each component often needs to be tailored for domain-specific datasets. Our approach focuses on ease of deployment and adaptability, where a multi-speaker ASR model is trained in the same way as mono-speaker ASR models, based on token objectives and cross-entropy loss.

## 3. Proposed Approach: Sortformer

### 3.1. Permutation Problem in Diarization

Speaker diarization or speaker-attributed ASR always accompanies issues of permutation matching between inferred speaker and the ground-truth speaker during training for calculating losses or evaluation processes to find the right speaker mapping. To tackle this issue, the concept of PIL or PIT was first popularized by the two studies (Kolbæk et al., 2017; Yu et al., 2017b) for the task of speech source separation, which inevitably requires the model to handle PIL calculation. Following this, (Fujita et al., 2019) adopted the concept of PIL for the task of speaker diarization, and later improved it in (Horiguchi et al., 2022a) by employing a sequence-to-sequence model to generate the attractors by training the model with PIL.

While PIL shows promising results in the aforementioned tasks, PIL-based end-to-end speaker diarization model is more challenging to integrate into ASR systems. Since PIL requires a specialized loss function at the model's output layer, it limits its applicability when training multi-speaker ASR models for multiple tasks simultaneously using the same ground truth. For example, if a model is trained for tasks like speech summarization, speech translation, and multi-speaker ASR concurrently, this constraint mandates a specialized loss calculation mechanism specifically designed for the multi-speaker ASR task. In contrast, the sorting-based approach does not impose special requirements on the loss function. Once the speaker tokens in the ground truth labels are sorted, the model can be trained using the standard cross-entropy function on text tokens (see Fig. 2). This approach improves ease of use and adaptability, especially for those unfamiliar with complex model architectures.

### 3.2. Diarization Model as a Multi-label Binary Classifier

We propose a model designed for the simultaneous estimation of class presences from a sequence of input tokens while the class labels follow the arrival time of each speaker's first segment. Consider a set of frame-wise $D$-dimensional embedding vectors, $\{\mathbf{x}_t\}_{t=1}^{T}$, where $\mathbf{x}_t \in \mathbb{R}^D, t = 1, 2, ..., T$ represents the frame index. Given the input sequence, the model is expected to generate the class presence vector sequence $\{\xi_t\}_{t=1}^{T}$, where $\xi_t \in \mathbb{R}^K, t = 1, 2, ..., T$. In this context, $\xi_t = [y_{1,t}, y_{2,t}, ..., y_{K,t}]^\top$ denotes the class presences of $K$ classes ($K$ potential speakers) at time $t$, where $y_{k,t} \in \{0, 1\}$ indicates the speech activity of the $k$-th speaker at the $t$-th frame.

$$P(\xi_1,...,\xi_T \mid \mathbf{x}_1,...,\mathbf{x}_T) = \prod_{k=1}^{K} \prod_{t=1}^{T} P(y_{k,t} \mid \mathbf{x}_1,...,\mathbf{x}_T) \quad (1)$$

Sortformer assumes the conditional independence of $y_{k,t}$ given the embedding vectors (features). Therefore, Sortformer employs *Sigmoid* instead of *Softmax* unlike the activation function for the output layer in the Transformer encoder (Vaswani et al., 2017). This assumption is formalized as in Eq. (1).

Under this framework, the task is construed as a multi-label classification problem, which is amenable to modeling via a neural network, denoted by $f_\Theta$. The model is defined as follows:

$$\mathbf{P} = [\mathbf{p}_1, ..., \mathbf{p}_T] = f_\Theta(\mathbf{x}_1, ..., \mathbf{x}_T), \quad (2)$$

where $\mathbf{p}_t = [p_{1,t}, ..., p_{K,t}]^\top \in [0, 1]^K$ represents the posterior probabilities of the presence of $K$ classes at frame index $t$, $\mathbf{P}$ is a $K$ by $T$ matrix that contains the columns of $\mathbf{p}_t$ vectors and $f_\Theta$ represents the Sortformer model with a set of parameters $\Theta$. Each $\hat{y}_{k,t}$ is defined as a binarized

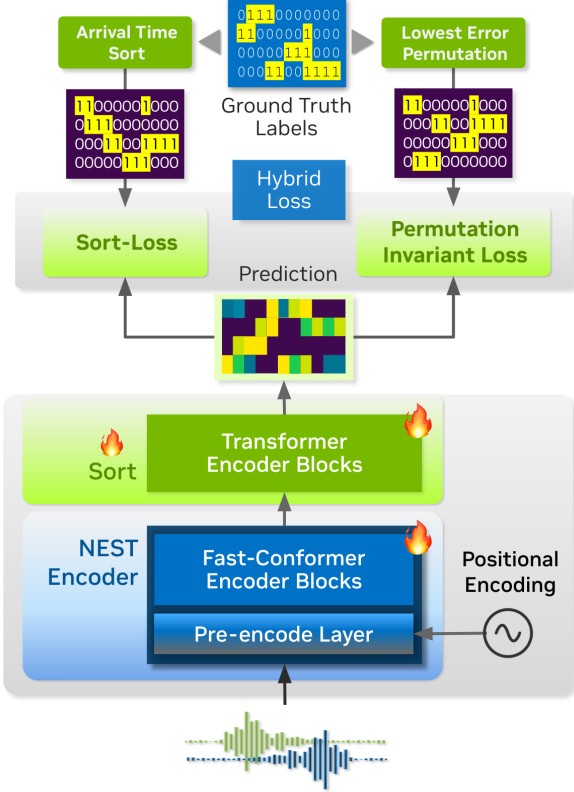

*Figure 3.* Sortformer architecture with hybrid loss.

value from $p_{k,t}$. Eq. (2) can be rewritten in matrix form by concatenating the vectors as $\mathbf{X} = [\mathbf{x}_1, \mathbf{x}_2, \ldots, \mathbf{x}_T]$, where each column of $\hat{\mathbf{Y}}$ represents the class presence at time $t$, i.e., $\hat{\mathbf{Y}} = [\hat{\xi}_1, \hat{\xi}_2, \ldots, \hat{\xi}_T]$, where $\{\hat{\xi}_t\}_{t=1}^T$ is the estimation of class presences.

If we decompose $\hat{\mathbf{Y}}$ in terms of speaker identity, the class presence matrix becomes $\hat{\mathbf{Y}} = [\hat{\mathbf{y}}_1, \hat{\mathbf{y}}_2, \ldots, \hat{\mathbf{y}}_K]^\top$, where $\hat{\mathbf{y}}_k$ is a row-wise speaker presence vector, $\hat{\mathbf{y}}_k = [\hat{y}_{k,1}, \hat{y}_{k,2}, \ldots, \hat{y}_{k,T}]^\top$, for the $k$-th speaker. Similarly, $\mathbf{P}$ can also be decomposed as $\mathbf{P} = [\mathbf{q}_1, \mathbf{q}_2, \ldots, \mathbf{q}_K]^\top$, where $\mathbf{q}_k$ is a row-wise speaker presence posterior probabilities, $\mathbf{q}_k = [p_{k,1}, p_{k,2}, \ldots, p_{k,T}]^\top$, for the $k$-th speaker. Thus, in its simplest form, the model can be represented as $\mathbf{P} = f_\Theta(\mathbf{X})$, where $\mathbf{X} \in \mathbb{R}^{D \times T}$ is the input sequence and $\mathbf{P} \in \mathbb{R}^{K \times T}$ is a matrix containing the speaker presence posterior probabilities.

### 3.3. Loss Calculation

**Binary Cross-Entropy**    The loss values for the individual sigmoid output $p_{k,t}$ in the aforementioned model, represented by $f_\Theta(\mathbf{X})$, are calculated using the binary cross-entropy (BCE) function. Let $p_{k,t}$ represent the class presence posterior probability in Eq. (2), where $p_{k,t} \in [0, 1]$. To simplify the notation, we drop $k$ and $t$. The BCE loss for a

single example is defined as:

$$\mathcal{L}_{\mathrm{BCE}}(y, p) = -\left(y \log(p) + (1 - y) \log(1 - p)\right), \quad (3)$$

where $y \in \{0, 1\}$ is the true speaker label for the example, and $p \in [0, 1]$ is the predicted speaker probability for the positive class.

**Permutation Invariant Loss**    Hereafter, we refer to the function that computes PIL as $\mathcal{L}_{PIL}$. The definition of PIL can be described as follows: Let $\mathbf{Y} = [\mathbf{y}_1, \ldots, \mathbf{y}_K]^\top \in \mathbb{R}^{K \times T}$ be the ground truth speaker presence matrix, and $\mathbf{P} = [\mathbf{q}_1, \mathbf{q}_2, \ldots, \mathbf{q}_K]^\top \in \mathbb{R}^{K \times T}$ be the predicted speaker presence matrix where $K$ denotes the number of speakers and $T$ denotes the number of frames. $\mathcal{L}_{PIL}$ aims to find the permutation $\pi$ that minimizes the error between the predicted matrix and the ground truth. Mathematically, it is defined as:

$$\mathcal{L}_{\mathrm{PIL}}(\mathbf{Y}, \mathbf{P}) = \min_{\pi \in \Pi} \left\{ \mathcal{L}_{\mathrm{BCE}}(\mathbf{Y}_\pi, \mathbf{P}) \right\}, \quad (4)$$

where $\Pi$ is the set of all possible permutations of the indices $\{1, \ldots, K\}$, and $\mathbf{Y}_\pi$ is the matrix $\mathbf{Y}$ permuted according to the permutation function $\pi$, i.e., $\mathbf{Y}_\pi = [\mathbf{y}_{\pi(1)}, \ldots, \mathbf{y}_{\pi(K)}]^\top$. If we express the Eq. (4) using the speaker-wise class presence vector $\mathbf{y}_k$ and speaker-wise posterior speaker probability $\mathbf{q}_k$, the equation becomes:

$$\mathcal{L}_{\mathrm{PIL}}(\mathbf{Y}, \mathbf{P}) = \min_{\pi \in \Pi} \left\{ \frac{1}{K} \sum_{k=1}^{K} \mathcal{L}_{\mathrm{BCE}}(\mathbf{y}_{\pi(k)}, \mathbf{q}_k) \right\} \quad (5)$$

$$= \min_{\pi \in \Pi} \left\{ \frac{1}{TK} \sum_{k=1}^{K} \sum_{t=1}^{T} \mathcal{L}_{\mathrm{BCE}}(y_{\pi(k),t}, p_{k,t}) \right\}. \quad (6)$$

**Sort Loss**    Sort Loss is designed to compare predicted outputs with true labels, typically sorted by arrival time order or another relevant metric. The key distinction Sortformer introduces compared to the previous end-to-end diarization systems such as EEND-SA (Fujita et al., 2019), EEND-EDA (Horiguchi et al., 2022a) lies in the organization of class presence matrix $\hat{\mathbf{Y}}$. Let $\Psi$ be a function that measures the arrival time of the first speaker segment for the corresponding speaker bin,

$$\Psi(\mathbf{y}_k) = \min\{t' \mid y_{k,t'} \neq 0, t' \in [1, T]\} = t_k^0 \quad (7)$$

where $t_k^0$ is the frame index of the first speaker segment for the $k$-th speaker. Sortformer is expected to generate values $\hat{\mathbf{y}}_k$ for each speaker index $k$, where the following condition holds:

$$\Psi(\hat{\mathbf{y}}_1) \leq \Psi(\hat{\mathbf{y}}_2) \leq \cdots \leq \Psi(\hat{\mathbf{y}}_k), \quad (8)$$

which indicates that the model function $f_\Theta$ learns to generate the class presence output $\hat{\mathbf{Y}}$ with row indices sorted in arrival time order. Let $\eta$ the sorting function applied

to the indices $\{1, \ldots, K\}$, and $\mathbf{Y}_\eta$ be the vector $\mathbf{y}$ sorted according to the arrival time order sorting function $\eta$, i.e.,

$$\eta(\mathbf{Y}) = \mathbf{Y}_\eta = \left(\mathbf{y}_{\eta(1)}, \ldots, \mathbf{y}_{\eta(K)}\right). \qquad (9)$$

Using the arrival time function defined in Eq. (7), accordingly, the following conditions hold in the ground truth vectors $\mathbf{y}_{\eta(k)}$ for all $K$ speakers:

$$\Psi(\mathbf{y}_{\eta(1)}) \leq \Psi(\mathbf{y}_{\eta(2)}) \leq \cdots \leq \Psi(\mathbf{y}_{\eta(K)}) \qquad (10)$$

Thus, sort-loss with the sorting function $\eta$ is defined mathematically as:

$$\mathcal{L}_{\text{Sort}}(\mathbf{Y}, \mathbf{P}) = \mathcal{L}_{\text{BCE}}(\mathbf{Y}_\eta, \mathbf{P}) = \frac{1}{K} \sum_{k=1}^{K} \mathcal{L}_{\text{BCE}}(\mathbf{y}_{\eta(k)}, \mathbf{q}_k), \quad (11)$$

where $\mathbf{y}_{\eta(\mathbf{k})}$ is the vector of true labels that are sorted in arrival time order resulting in the sorted index $\eta(k)$, $\mathbf{q}_k$ is the vector of predicted outputs, $\mathcal{L}_{\text{BCE}}(\mathbf{y}_{\eta(k)}, \mathbf{q}_k)$ represents the loss for the $k$-th speaker, and $K$ is the total number of speakers.

**Hybrid Loss**   While Sortformer can be trained solely with Sort Loss, there is a limitation that arrival time estimation is not always correct. This issue becomes more pronounced as the number of speakers increases during the training session. Note that Sortformer models can be trained using Sort Loss only, PIL only, or a hybrid loss by setting the weight between these two losses. The hybrid loss $\mathcal{L}_{\text{hybrid}}$ can be described as follows:

$$\mathcal{L}_{\text{hybrid}} = \alpha \cdot \mathcal{L}_{\text{Sort}} + (1 - \alpha) \cdot \mathcal{L}_{\text{PIL}}, \qquad (12)$$

where $\alpha$ is an empirically determined weighting factor.

### 3.4. Transformer Encoder Learns to Sort

Sort Loss, along with sorted target objectives, enables the model to learn the sorting of arrival times as it generates frame-level speaker labels. Therefore, a model trained with Sort Loss can be viewed as performing *neural sorting*, as the sorting operation is integrated into the Transformer's matrix multiplication process. Sortformer models are trained by minimizing the Sort Loss, which is used to train $f_\Theta$:

$$\mathcal{L}_{\text{Sort}}(\mathbf{Y}, f_\Theta(\mathbf{X})) = \mathcal{L}_{\text{BCE}}(\mathbf{Y}_\eta, f_\Theta(\mathbf{X})). \qquad (13)$$

It is worth noting that our model differs from conventional Transformer-based end-to-end diarization systems, such as EEND-SA (Fujita et al., 2019) and EEND-EDA (Horiguchi et al., 2022a) through its use of positional embeddings. These baseline systems do not require positional embeddings, as the ordering of speaker labels is not relevant to these end-to-end diarization models. However, the multi-head self-attention (MHA) in Transformers (Vaswani et al.,

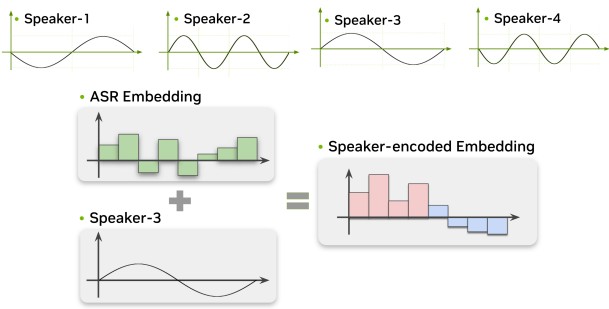

*Figure 4.* Sinusoidal kernels are applied to represent speaker supervisions on top of the ASR embeddings.

2017) inherently exhibits permutation equivariance when positional embeddings are omitted (see Appendix Sections E and F). Therefore, Sortformer employs positional embeddings to provide the model with a sense of sequence ordering.

## 4. Bridging Timestamps and Tokens

### 4.1. Resource-Efficient Training with Adapters

To effectively leverage the knowledge from a pretrained ASR model, we incorporate adapters for multi-speaker ASR tasks, as outlined in (Wang et al., 2024). A common challenge with fully fine-tuning a pretrained ASR model on new tasks is that it tends to forget previous tasks. In our case, the primary distinction between single-speaker and multi-speaker ASR lies in the insertion of speaker tokens into the single-speaker transcripts. Consequently, preserving the previously acquired knowledge becomes crucial for multi-speaker ASR. This makes the use of adapters, as described in (Houlsby et al., 2019), a more effective approach.

### 4.2. Speaker Supervision with Speaker Kernel

The most crucial part of integrating the speaker diarization model and the ASR model is how the words or tokens are assigned to speaker labels. In our framework, the diarization result is treated as a speaker encoding by injecting information through differentiable kernels. Fig. 4 shows how sinusoidal kernels are added to the original ASR encoder states. Let $\gamma_k$ be the speaker kernel for the $k$-th speaker. We define $\kappa_{k,z} = \sin((2\pi kz)/M)$, $\gamma_k = \left[\kappa_{k,1}, \kappa_{k,2}, \ldots, \kappa_{k,M}\right]$, and $\Gamma = [\gamma_1, \gamma_2, \ldots, \gamma_K]^\top$, where $M$ is the dimension of the ASR encoder state, $z$ is the embedding vector bin index, and $\Gamma \in \mathbb{R}^{K \times M}$ contains the kernels for $K$ speakers. We employ additive kernels based on the aforementioned sinusoidal functions. The following equation represents the kernel-based speaker encoding:

$$\tilde{\mathbf{A}} = \frac{\mathbf{A}}{\|\mathbf{A}\|_2} + \mathbf{\Gamma}^T \cdot \mathbf{P}, \qquad (14)$$

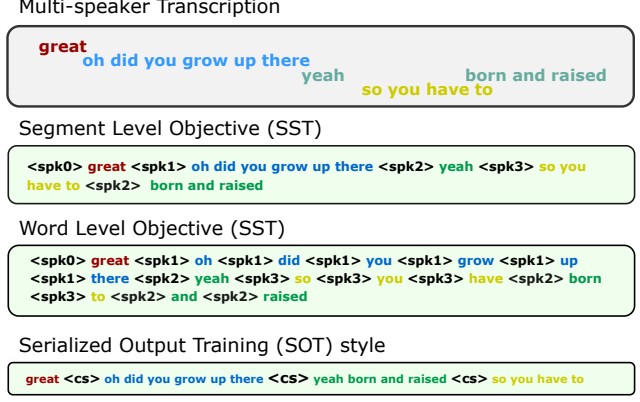

Multi-speaker Transcription

Segment Level Objective (SST)

Word Level Objective (SST)

Serialized Output Training (SOT) style

*Figure 5.* Three types of transcriptions for multi-speaker ASR model training.

where $\mathbf{A}$ is the encoder state (also referred to as the ASR embedding) from the encoder part of the ASR model, and $\|\mathbf{A}\|_2$ denotes the L2 norm of each column (feature vector) in $\mathbf{A}$. $\tilde{\mathbf{A}}$ is the speaker-encoded encoder state matrix with dimensions $M$ by $T$. $\mathbf{P}$ is the output from Eq. (2), which we refer to as *speaker supervision*.

### 4.3. Sorted Speaker Token-Objectives in Transcript

**Sorted Serialized Transcript**    We use speaker tokens that represent the generic speaker labels such as <spk0>, <spk1>, $\cdots$ <spkK>. These speaker tokens appear as single tokens in both predicted text and ground truth text. In the ground truth text, these tokens are also sorted in arrival time order, meaning the first appearing speaker is assigned <spk0> where the second appearing speaker is assigned <spk1> and so on. Therefore, if both a word and the corresponding speaker's speech segment are recognized correctly, these speaker tokens and speaker kernels are aligned by the decoder. As a result, the ASR model and Sortformer diarization model can be trained or fine-tuned using a standard cross-entropy loss on these sorted tokens (including speaker tokens), without requiring a separate PIT or PIL mechanism for the ASR decoding stage. We refer to the transcriptions that include sorted word-level objectives as *Sorted Serialized Transcript* (SST). Our approach does not require word-level timestamps for training, thanks to the word-timestamp approximation scheme (see Appendix C for more details).

**Word level vs. Segment level**    In our proposed framework for training multi-speaker ASR models, speaker tokens can be applied at two different levels as described in Figure 5. A speaker token is placed in front of every word. The order of words is determined by comparing the onset (start time) of each word. The segment-level objective is conceptually similar to the SOT  (Kanda et al., 2020b; 2021) style, while the speaker tokens used in our work are sorted speaker indices and not the change of speaker token. In comparison,

SOT focuses on speaker change point token <cs> with serialized outputs, while SST does not employ speaker change points but employs sorted speaker tokens to assign a generic speaker index (*e.g.*, <spk3>) for each word. On the other hand, the word-level objective simply places speaker tokens before each and every word.

**Permutation Resolution via Sorting**    In this work, all multi-speaker ASR training sessions use the same cross-entropy loss as conventional single-speaker ASR models, without relying on permutation-invariant or alternative permutation-handling losses. Instead, the permutation problem is resolved by directly matching the model's logit outputs with speaker and word tokens that are sorted by speaker arrival time, as illustrated in Figure 2 and Figure 5. Speaker tokens in the ground truth transcriptions are sorted by arrival time during data preparation, and the Sortformer module is designed to generate speaker predictions in the same arrival-time order, ensuring correct alignment with these ground truth labels. During the multi-speaker ASR training, Sortformer can either be fine-tuned, or its weights can be kept frozen. Regardless of whether Sortformer's weights are fine-tuned or frozen, the training of the multi-speaker ASR system is governed solely by the cross-entropy loss applied to the output tokens.

## 5. Experimental Results

### 5.1. Diarization Model Training

#### 5.1.1. DATASETS

For training data of the Sortformer end-to-end diarizer, we use a combination of 2,030 hours of real data (Fisher English Training Speech Part1 and 2 (Cieri et al., 2004), AMI Corpus Individual Headset Mix (IHM) (Kraaij et al., 2005) using the train and dev split from (Landini et al., 2022), DIHARD3-dev (Ryant et al., 2020), VoxConverse-v0.3 (Chung et al., 2020), ICSI (Janin et al., 2003), AISHELL-4 (Fu et al., 2021), NIST SRE 2000 CALLHOME Part1 [2] (Przybocki & Martin, 2001) which we refer to as CALLHOME) and 5150 hours of audio mixture data (created using using LibriSpeech (Panayotov et al., 2015) and NIST SRE04-10 (Doddington et al., 2000; Gonzalez-Rodriguez, 2014) as source datasets) generated by an open-source speech data simulator (Park et al., 2023).

All parameters for audio mixture generated using an open-source speech data simulator (Park et al., 2023) are default settings except that the overlap ratio is set to 0.12 and the average silence ratio is set to 0.1. We evaluate the model performance on DIHARD3-eval (Ryant et al., 2020) (referred to

---

[2]We use the two-fold splits from the Kaldi x-vector recipe (Przybocki & Martin, 2001) where `Part1` is used for training and fine-tuning and `Part2` for evaluation

*Table 1.* DER results on speaker diarization. All evaluations include overlapping speech. Dataset name, number of speakers, and collar length are shown. Underlined values are the best-performing Sortformer evaluations. A single Sortformer model is trained for each loss type and evaluated on three datasets. Systems marked with a cross (†) involve a clustering phase and are not strictly end-to-end.

| Diarization Systems | Model Size | Post Proc. | **DH3** $n_{\mathrm{Spk}} \leq 4$, 0.0 s | **CALLHOME-part2** $n_{\mathrm{Spk}}=2$, 0.25 s | $n_{\mathrm{Spk}}=3$, 0.25 s | $n_{\mathrm{Spk}}=4$, 0.25 s | **CH109** $n_{\mathrm{Spk}}=2$, 0.25 s |
|---|---|---|---|---|---|---|---|
| (Park et al., 2022) †MSDD | 31.1M | - | 29.40 | 11.41 | 16.45 | 19.49 | 8.24 |
| (Horiguchi et al., 2022a;b) EEND-EDA | 6.4M | - | 15.55 | 7.83 | 12.29 | 17.59 | - |
| (Chen et al., 2022) †WavLM-L+EEND-VC | 317M | - | - | 6.46 | 10.69 | **11.84** | - |
| (Horiguchi et al., 2022b) †EEND-GLA-Large | 10.7M | - | **13.64** | 7.11 | 11.88 | 14.37 | - |
| (Chen et al., 2024) AED-EEND | 11.6M | - | - | 6.18 | 11.51 | 18.44 | - |
| (Chen et al., 2024) AED-EEND-EE | 11.6M | - | - | 6.93 | 11.92 | 17.12 | - |
| Sortformer-PIL | 123M | ✗ | 18.33 | 7.28 | 11.57 | 18.80 | **5.66** |
| | | ✓ | 17.04 | 6.94 | 10.30 | 17.52 | 6.89 |
| Sortformer-Sort-Loss | 123M | ✗ | 17.88 | 7.42 | 12.68 | 19.42 | 9.08 |
| | | ✓ | 17.10 | 6.52 | 10.36 | 17.40 | 10.85 |
| Sortformer-Hybrid-Loss | 123M | ✗ | 16.28 | 6.49 | 10.01 | 14.14 | 6.27 |
| | | ✓ | 14.76 | **5.87** | **8.46** | 12.59 | 6.86 |

as DH3 in Table 1), CALLHOME-part2 (Przybocki & Martin, 2001), and a two-speaker subset of 109 sessions from Callhome American English Speech (CHAES) (Canavan et al., 1997), which we refer to as CH109. In DIHARD3-eval, we include only sessions with four or fewer speakers.

### 5.1.2. DATA CLEANING

For the Fisher English Training Speech (Cieri et al., 2004), AMI (Kraaij et al., 2005), and NIST SRE 04-10 datasets (Doddington et al., 2000; Gonzalez-Rodriguez, 2014), we refined the speaker annotations by applying a multilingual speech activity detection (SAD) model from an open-source toolkit and a pretrained Sortformer diarizer model to gain more accurate and tight boundaries where the minimal amount of silence exists between the onset and offset of speech and the segment start and end. For datasets such as AMI (Kraaij et al., 2005), ICSI (Janin et al., 2003), AISHELL-4 (Fu et al., 2021) where more than four speakers exist and/or session lengths are far longer than the 90-second limit, we truncated the dataset into 90-second short segments and retained only those segments containing less than or equal to four speakers.

### 5.1.3. TRAINING SETUP

Our model is based on the L-size NEST (Huang et al., 2025) encoder (115M parameters). We use 18 layers of Transformer (Vaswani et al., 2017) encoder blocks with a hidden size of 192, and two feed-forward layers with four sigmoid outputs on top of it (See Fig. 3). In total, including the NEST encoder, we evaluate a 123M parameter Sortformer model. We employ a two-stage training strategy on the Sortformer model: pretraining stage with both real and simulated data, and fine-tuning stage with real data

only. We use 90-second long training samples and a batch size of 4. We use *adamW* (Loshchilov, 2017) optimizer with a learning rate of $10^{-4}$ and a weight decay of $10^{-3}$. The minimum learning rate is $10^{-6}$. We use 2,500 steps of warmup where inverse square-root annealing is employed for learning-rate scheduling. A dropout rate of 0.5 is used for Transformer encoder layers and feedforward layers, and 0.1 is used for NEST encoders. We do not employ any special augmentation schemes such as SpecAugment (Park et al., 2019). All Sortformer models are trained on 8 nodes of $8\times$NVIDIA Tesla V100 GPUs. Relative to pure Permutation Invariant Loss (PIL) training, which has an average epoch time of 1,020 seconds, our sorting-based approach introduces minimal training time overhead. The average epoch duration increases by 0.22% to 1,022.28 seconds for a pure sort loss configuration, and increases by 2.26% to 1,043.1 seconds for a hybrid loss setup.

### 5.2. Results on Speaker Diarization Task

Table 1 shows the experimental results of diarization evaluation on Sortformer diarizer. We evaluate three models trained with three different loss types: PIL only, Sort Loss only, and hybrid loss with $\alpha = 0.5$ in Eq. (12). We train the Sortformer model to handle up to 4 speakers, so we compare the popular neural diarizers that are reporting speaker-wise diarization error rate (DER) on each dataset. In addition, it is crucial to remind that Sortformer is not individually fine-tuned on three evaluation datasets, unlike the systems in EEND-EDA (Horiguchi et al., 2022a), EEND-GLA (Horiguchi et al., 2022b) and WavLM+EEND-VC (Chen et al., 2022). We apply timestamp postprocessing that mitigates the errors generated from collar length and annotation style of the datasets. See Appendix B for details of the postprocessing. A noteworthy result from Table 1 is

*Table 2.* Evaluation results of *Sortformer-MS-Canary* on short segments from AMI test and CH109. Underlined numbers are the best performing setups without adapter. Except the baseline, the ASR encoder and decoder are fine-tuned in all systems.

| System Index | Obj. Level | Model Param. Size | Train Speaker Supervision | Infer Speaker Supervision | Diar. Model Fine-tune | Adapter Dim. | AMI-test ($\leq$ 4-spks) WER | cpWER | CH109 (2-spks) WER | cpWER |
|---|---|---|---|---|---|---|---|---|---|---|
| baseline | - | 170M | - | - | - | - | 26.93% | - | 21.81% | - |
| 1 | word | 170M | - | - | - | - | 19.67% | 32.94% | 18.57% | 24.80% |
| 2 | word | 293M | Sortformer | Sortformer | ✗ | - | 20.08% | 28.17% | 18.65% | 22.22% |
| 3 | word | 293M | Sortformer | Sortformer | ✓ | - | 19.47% | 32.74% | 19.53% | 26.97% |
| 4 | word | 293M | Ground Truth | Sortformer | - | - | 19.48% | 26.83% | 18.74% | 24.39% |
| 5 | segment | 1.12B | Sortformer | Sortformer | ✗ | 256 | 18.58% | 28.59% | 17.74% | 22.19% |
| 6 | word | 1.12B | Sortformer | Sortformer | ✗ | 256 | **18.04%** | **26.71%** | **16.46%** | **21.45%** |

*Table 3.* Comparison of WER on the LibriSpeechMix dataset. Evaluations marked with a cross (✝) are tested on audio mixtures with a fixed delay for each speaker in the dataset.

| ASR Systems | Param. Size | Spk. Spv. | WER 1mix | 2mix | 3mix |
|---|---|---|---|---|---|
| Canary ASR | 170M | ✗ | 2.19 | 21.37 | 48.71 |
| (Puvvada et al., 2024) | 1B | ✗ | **1.65** | 20.49 | 47.32 |
| SOT-ASR | 135.6M | ✗ | 4.6 | 11.2 | 24.0 |
| (Kanda et al., 2020b) | | | | | |
| SOT-ASR-SQR | 135.6M | ✗ | 4.2 | 8.7 | 20.2 |
| (Kanda et al., 2020a) | | | | | |
| DOM-SOT | 33M | ✗ | 5.17 | 5.56✝ | 9.96✝ |
| (Shi et al., 2024) | | | | | |
| MT-LLM | 8.4B | ✓ | 2.3 | 5.2 | 10.2 |
| (Meng et al., 2025) | | | | | |
| MS-Canary | 170M | ✗ | 2.74 | 6.55 | 12.14 |
| **Sortformer-MS-Canary** | 293M | ✓ | 2.26 | **4.61** | **9.05** |

that Sort Loss alone achieves performance comparable to that of the traditional PIL-trained model. Because Sort Loss offers a competitive training signal, combining it with PIL in a hybrid loss allows the model to leverage strengths from both, leading to performance that surpasses models trained with either loss alone.

## 5.3. Multi-speaker ASR Training Data

### 5.3.1. DATASETS

The training dataset used for real-life multi-speaker recording experiments includes the AMI (Kraaij et al., 2005) Individual Headset Mix (IHM) train split, which has been used in previous research. It also includes the ICSI (Janin et al., 2003) dataset, the DipCo dataset (Segbroeck et al., 2020), and a 30K segment subset of the Fisher English Training Speech Part 1 and 2 dataset. The first three sets collectively contain 138 hours of multi-speaker speech, with up to four speakers per sample. The Fisher dataset comprises 2,000 hours of two-speaker data. To address the speaker data imbalance, 30K samples are randomly selected from the Fisher dataset and incorporated into our four-speaker data blend. The resulting combined training corpus consists of 230 hours of multi-speaker audio, with a maximum of four speakers per sample. We report word error rate (WER) and concatenated minimum-permutation

WER (cpWER) (Watanabe et al., 2020) for real-life multi-speaker recording experiments. See Appendix D for detailed descriptions of the evaluation metrics.

For comparative studies, we evaluate our model on Lib-riSpeechMix (Kanda et al., 2020b), which is the most popular artificial audio mixture dataset for testing harsh overlap speech for multi-speaker ASR systems. We follow the train, validation, and test set split as described in (Kanda et al., 2020b) and created 2M audio mixtures from the LibriSpeech (Panayotov et al., 2015) corpus. For the LibriSpeechMix dataset, WER is reported following the methodology described in (Kanda et al., 2020b;a).

### 5.3.2. TRAINING SETUP

For the real-life multi-speaker recording experiments, we build upon the Canary architecture (Puvvada et al., 2024), extending its capabilities to process multi-speaker input. We use the 170M variant for fine-tuning experiments and the 1B model for our adapter experiments following the adapter approach outlined in (Wang et al., 2024). For all these experiments, a single NVIDIA RTX 6000 Ada GPU is used.

We train the Canary-170M models for 50K steps on the multi-speaker ASR training data blend, using a batch size of 64. Both the Fast-Conformer encoder and Transformer decoder parameters are fully fine-tuned. Speaker information is integrated through the Sortformer model, whose output is combined with the ASR encoder embedding via a sinusoidal kernel. For the Canary-1B model, all other model parameters are frozen, and only the adapter parameters in the encoder and decoder are learned over 75K updates on the multi-speaker ASR training data blend, starting from random initialization. All models are trained using the AdamW (Loshchilov, 2017) optimizer, with a weight decay of $10^{-3}$, inverse square root annealing, a warm-up of 2,500 steps, a peak learning rate of $3.10^{-4}$, and a minimum learning rate of $10^{-6}$.

For the LibriSpeechMix experiments, we use System 2 from Table 2 with the 170M ASR model, the model without an adapter while using the SOT-style speaker token objective to test the effectiveness of the SOT approach. First, we fine-

tune the Sortformer model from the evaluation in Table 2 on the 960-hour LibriSpeechMix training dataset. Then we run 180K steps of fine-tuning of the ASR model while keeping the Sortformer model frozen, obtaining the *Sortformer-MS-Canary* model. As a baseline, we also fine-tune the Canary-170M model on LibriSpeechMix data for the same number of updates without any speaker supervision from Sortformer. This model is referred to as MS-Canary in Table 3.

## 5.4. Results on Multi-speaker ASR

### 5.4.1. ABLATION STUDY DESIGN

We perform an ablation study on real-life multi-speaker recordings to gauge the contribution of each component. As a baseline system, we use the Canary-170M (Puvvada et al., 2024) ASR model in its original form without any fine-tuning. Table 2 shows the various setups we evaluate to show the contributions of each component. The baseline system is a single-speaker Canary-170M (Puvvada et al., 2024) model that is not trained on the multi-speaker ASR dataset. The original Canary-170M model does not have speaker tokens; therefore, cpWER is not calculated. See Appendix D for detailed description about WER calculation. The baseline system shows how challenging the evaluation set is for the vanilla ASR model. System 1 is the most primitive model where neither speaker supervision nor adapters are used. System 2 and System 3 are the models where Sortformer diarization module is plugged in while Sortformer model weights are frozen in System 2 and fine-tuned in System 3. Finally, System 4 is a system trained with ground-truth speaker labels fed through a speaker kernel but Sortformer is used as speaker supervision during inference. System 5 and System 6 are the multi-speaker ASR models trained with the adapter technique in (Wang et al., 2024), with the Canary-1B model. Systems 5 and 6 show the best-performing setup in both segment-level and word-level objectives. However, System 5 not only shows degradation in segment-level objectives, but we also observe this decline across all types of settings and datasets.

### 5.4.2. COMPARATIVE EVALUATION

For evaluation on the LibriSpeechMix benchmark, shown in Table 3, we compare our proposed model with other top-performing systems in the literature that report WERs across all three mixture sets with the same model. See Appendix D for the WER calculation in Table 3. *Sortformer-MS-Canary* achieves the best performance on multi-talker datasets (2-mix and 3-mix), while the baseline single-speaker Canary ASR models (170M and 1B) show slightly better performance on 1-mix (single speaker). Furthermore, the improvement from MS-Canary to *Sortformer-MS-Canary* indicates that we can successfully integrate a 123M-parameter Sortformer model, yielding a relative error rate reduction of

30% for 2-mix and 25% for 3-mix, while showing minor degradation on single-speaker 1-mix audio when compared to the baseline Canary-170M model.

### 5.4.3. RUNTIME PERFORMANCE

Runtime evaluations were performed on a single NVIDIA RTX A6000 Ada GPU. The stand-alone Sortformer diarization model was benchmarked on the LibriSpeechMix test-3mix dataset (42,514.9s total audio, using 10-run averages). In multi-speaker ASR tasks (batch size: 100), integrating Sortformer supervision with the MS-Canary system (170M parameters) increased processing time by a mere 0.78%, increasing from 297.891s to 300.213s for the Sortformer-MS-Canary system (293M parameters).

## 6. Conclusion

In this paper, we propose Sortformer, an encoder-type diarization model designed for integration with speech-to-text systems. By learning an arrival-time sorting mechanism, Sortformer enables permutation-resolved speaker supervision, thereby supporting cross-entropy loss-based training and unifying multi-speaker ASR frameworks with the principles of monaural ASR frameworks. In addition, we show that combining Sort Loss with PIL as a hybrid loss improves stand-alone diarization performance when trained solely on PIL. Finally, we demonstrate that the proposed framework can improve multi-talker ASR benchmarks to a system without speaker supervision. Future work will explore streaming systems, target-speaker ASR features, and multi-task capabilities such as translation and summarization. We hope our proposed work serves as an accessible baseline to inspire further research in multi-speaker ASR.

## Impact Statement

This research introduces Sortformer, a novel encoder-based diarization model that addresses the challenging speaker permutation problem in multi-speaker STT systems. By employing a Sort Loss function based on speaker arrival times, Sortformer enables permutation-resolved speaker supervision. This innovation streamlines the integration of speaker tagging, allowing STT models to be trained with standard cross-entropy loss, similar to simpler mono-speaker systems, and reduces complex annotation needs. Experiments confirm improved diarization performance and transcription accuracy. The broader impact lies in making advanced multi-speaker ASR more accessible and easier to deploy. We foresee Sortformer accelerating the development of more versatile and accurate speaker-aware applications, paving the way for seamless integration into foundational STT models and LLMs, thus benefiting a wide range of interactive technologies.

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

## A. Data Cleaning

In our proposed multi-speaker ASR model, speaker tokens are generated by the model to predict the corresponding speaker label for each word. During training, we clean the data according to the following rules:

- We segment the long-form audio into shorter segments, each ranging between 10 and 20 seconds.

- Words in the transcripts are sorted based on their arrival time, even when they overlap, as illustrated in Figure 5. Overlapping speech results in more frequent speaker changes.

- If word-level timestamps are missing, we simulate them from segment-level timestamps. Specifically, we split the words into syllables and assume that each syllable has the same duration. The timestamp of each word is then estimated based on the number of syllables of each word and the average duration of each syllable.

- Speaker tokens are assigned based on the arrival time of each speaker, starting from <spk0>, followed by <spk1>, <spk2>, and so on.

- Samples with more than a 1-second overlap at the beginning or end are excluded.

- Samples where the first speaker only has one or two filler words at the beginning are excluded.

## B. Postprocessing of Speaker Diarization Segments (Timestamps)

We apply timestamp postprocessing that mitigates the errors generated from collar length and annotation style differences from multiple datasets. Our postprocessing step consists of:

1. **Onset threshold**: The threshold for detecting the beginning of speech.

2. **Offset threshold**: The threshold for detecting the end of speech.

3. **Onset padding**: The duration added at the beginning of each speech segment.

4. **Offset padding**: The duration added at the end of each speech segment.

5. **Minimum duration (on)**: The minimum duration required to retain a speech segment, used to remove short non-speech segments.

6. **Minimum duration (off)**: The minimum duration required to retain a non-speech segment, used to remove very short speech segments.

The parameters are tuned on two different splits of datasets: Set-A on DIHARD3 (Ryant et al., 2020) Dev split and Set-B on CALLHOME Part1 (Przybocki & Martin, 2001). Then Set-A parameters are applied to DIHARD3-eval, and Set-B is applied to CALLHOME Part2 and CH109. This speaker diarization postprocessing scheme is inspired by the postprocessing procedure in (Medennikov et al., 2020a). We optimize these floating-point postprocessing parameters with Optuna(Akiba et al., 2019) software.

## C. Word Timestamp Approximation

We employ a syllable-based word timestamp approximation technique for word-level objectives. After end-to-end ASR models gained popularity, such as RNNT-based models and attention encoder-decoder (AED) models, the ASR training process no longer requires word-by-word timestamp (alignment of words). Thus, securing speech datasets with word timestamps is difficult, and it becomes even more challenging when it comes to multi-speaker conversations because overlaps make it hard to be aligned with the forced aligners. Therefore, we propose a method to train a model without providing the model with timestamps by approximating the timestamps:

$$\ell = t_{\text{end}} - t_{\text{start}} \tag{15}$$

$$\tau_i^{\text{word}} = \left[ \delta_i, \delta_i + \frac{\ell}{N} n \right] = [\delta_i, \delta_i + \lambda n,] \tag{16}$$

- Original Segment-level Transcript

> **California does experience rain, but it's not always frequent.**
>
> **yeah** **that's right**

- Syllable-level Break Down

> cal | i | for | nia | does | ex | pe | ri | ence | rain | but | It's | not | al | ways | fre | quent
>
> yeah | that's | right

- Sorted Serialized Transcript

> <spk0> california <spk0> does <spk0> experience <spk1> yeah <spk0> rain
> <spk0> but <spk0> not <spk1> that's <spk0> always <spk1> right <spk0> frequent

*Figure 6.* Process of generating pseudo word timestamp and sorted serialized transcript.

where N is the total number of syllables in a segment, $\delta_i$ is the start time of the $i$-th word, $t_{\text{start}}$ and $t_{\text{end}}$ are start and end time of the segment. $\ell$ represents the segment length, $\lambda = \frac{\ell}{N}$ denotes the average syllable duration (speaking rate), defined as the segment length divided by the total number of syllables within that segment. This rate, denoted by $\lambda$, provides an average measure of how quickly syllables are spoken during the segment. Hence, $\lambda$ value is used to normalize the segment length and derive the start time and end time of each word. $\tau_i^{\text{word}}$ represents the timestamp of the $i$-th word, and $n$ denotes the number of syllables in the $i$-th word.

Figure 6 shows how word-timestamps are calculated in the absence of word-by-word timestamps. We split the words into syllable levels and assume that each syllable has the same duration. Thus, the timestamp of each word is then estimated based on the number of syllables of each word and the average duration of each syllable. The proposed word-timestamp approximation ensures the approximated word timestamps are comparable to the original word timestamps.

## D. Word Error Rate (WER) Calculation

When measuring the accuracy of multi-speaker ASR models, we need to assess both speaker tagging accuracy and word error rate (WER) itself, which comprises insertion, deletion, and substitution errors. However, in some of our experiments, we evaluate monaural ASR on multi-speaker recordings or artificial audio mixtures. Additionally, in Table 3, we report multi-speaker ASR system results using cpWER (Watanabe et al., 2020) for real-life multi-speaker recordings, as well as a method referred to as WER in previous studies(Kanda et al., 2020b;a) that introduced the LibriSpeechMix dataset. Furthermore, we report WERs on the monaural ASR systems tested on multi-speaker datasets. The WER values reported in the literature (Shi et al., 2024; Meng et al., 2025) could lead to misconceptions, as different groups of authors exhibit discrepancies in their descriptions. Here, we clarify the schemes we use to calculate WER values.

1. **cpWER** (**C**oncatenated Minimum-**P**ermutation **WER**): cpWER multi-speaker ASR on multi-speaker recordings: The cpWER metric evaluates speech recognition and diarization jointly through the following three steps:

   - **Concatenation**: Merging all utterances per speaker in both the reference and hypothesis.
   - **Permutation Scoring**: Computing WER across all possible speaker permutations (e.g., 24 permutations for 4 speakers).
   - **Optimal Selection**: Selecting the permutation with the lowest WER as the final score.

   As proposed in (Watanabe et al., 2020), cpWER inherently captures diarization errors and is used as the primary evaluation metric. Additionally, utterance-level error breakdowns are reported for detailed analysis.

2. **WER for monaural ASR on multi-speaker recordings** (in Table 2): For the baseline model, Canary 170M, we use word timestamps from annotations or forced-alignment results to sort the words based on their start times. We then evaluate WER in the same manner as standard monaural ASR evaluation.

3. **WER for multi-speaker ASR on multi-speaker recordings** (in Table 2): For all systems in Table 2, we remove the speaker tokens from both hypothesis and reference and measure WER values from the speaker-token removed hypothesis and reference pairs.

4. **WER of monaural ASR for the LibriSpeechMix dataset** (in Table 3): Baseline Canary ASR models (170M and 1B) fall into this category. We use the optimal reference combination (ORC) WER from the open-source toolkit (von Neumann et al., 2023). In ORC WER, speaker labels are ignored, and WER is measured based solely on hypothesis and reference transcript pairs. Although speaker labels are ignored, WER in SOT approaches still reflects speaker tagging accuracy, since SOT concatenates all word outputs for each speaker.

5. **WER of Multi-Speaker ASR for the LibriSpeechMix Dataset** (in Table 3): The multi-speaker (MS) version of Canary falls into this category. We use cpWER from the open-source toolkit (von Neumann et al., 2023). This aligns with the WER definitions in (Kanda et al., 2020a; Shi et al., 2024), where a mapping that delivers the lowest WER between predicted speakers and the ground-truth speakers is found, and then the WERs for all speakers are summed.

# E. Permutation Properties

## E.1. Definitions

Here are definitions and properties needed for clearly describing the permutation equivariance in the multi-head self-attention (MHA) mechanism in Transformer architectures. Note that we assume no positional embeddings are used on any input matrix or embedding.

**Definition E.1** (Permutation Function $\pi$). Let $n \in \mathbb{Z}_+$ be a positive integer. A permutation is defined as any bijective transformation of the finite set $\{1, \ldots, n\}$ into itself. Thus, a permutation is a function $\pi : \{1, 2, \ldots, n\} \longrightarrow \{1, 2, \ldots, n\}$ such that, for every integer $i \in \{1, \ldots, n\}$, there exists exactly one integer $j \in \{1, \ldots, n\}$ for which

$$\pi(j) = i. \tag{17}$$

**Definition E.2** (Permutation Matrix $P_\pi$). Let $S = \{1, 2, \ldots, n\}$ be a finite set of $n$ integers. A permutation $\pi$ is a bijective function from $S$ to itself, $\pi : S \to S$. The permutation matrix $P_\pi \in \mathbb{R}^{n \times n}$ corresponding to the permutation $\pi$ is defined such that its entry in the $i$-th row and $j$-th column, denoted as $(P_\pi)_{ij}$, is given by:

$$(P_\pi)_{ij} = \begin{cases} 1 & \text{if } i = \pi(j) \\ 0 & \text{otherwise} \end{cases} \tag{18}$$

This can also be written using the Kronecker delta symbol as follows:

$$(P_\pi)_{ij} = \delta_{i, \pi(j)} \tag{19}$$

The permutation matrix $P_\pi$ can be expressed by employing standard basis vectors. Let $e_k$ be the $k$-th standard basis vector in $\mathbb{R}^n$ (a column vector with a 1 in the $k$-th position and 0s elsewhere). The permutation matrix $P_\pi$ can be constructed by arranging the standard basis vectors $e_{\pi(j)}$ as its columns:

$$P_\pi = \begin{bmatrix} | & | & & | \\ e_{\pi(1)} & e_{\pi(2)} & \cdots & e_{\pi(n)} \\ | & | & & | \end{bmatrix} \tag{20}$$

This means the $j$-th column of $P_\pi$ is the vector $e_{\pi(j)}$.

**Definition E.3** (Spatial Permutation). Given a spatial permutation $\pi$, the transformation $T_\pi$ of a feature map $\mathbf{X} \in \mathbb{R}^{n \times d}$ is given by:

$$T_\pi(\mathbf{X}) = P_\pi \mathbf{X} \tag{21}$$

where $P_\pi \in \mathbb{R}^{n \times n}$ is the permutation matrix.

### E.2. Properties

*Property* 1 (*Permutation Invariance*). Let $\mathcal{X}$ be a set and $\mathcal{Y}$ be a codomain. A function $f : 2^{\mathcal{X}} \to \mathcal{Y}$ is said to be *permutation invariant* if, for any subset $\{x_1, ..., x_M\} \subseteq \mathcal{X}$ and any permutation $\pi$ of the indices $\{1, ..., M\}$, the following holds:

$$f(x_1, ..., x_M) = f\left(x_{\pi(1)}, ..., x_{\pi(M)}\right). \tag{22}$$

This property means that the output of $f$ is independent of the order of the elements in its input set. In a matrix form, an operator $A : \mathbb{R}^{d \times n} \to \mathbb{R}^{d \times n}$ is spatially permutation invariant if:

$$A(T_\pi(\mathbf{X})) = A(\mathbf{X}), \tag{23}$$

for any input $\mathbf{X}$ and any spatial permutation $\pi$.

*Property* 2 (*Permutation Equivariance*). Let $\pi$ be a permutation of $\{1, 2, ..., n\}$, and let $f : \mathbb{R}^n \to \mathbb{R}^m$ be a function. Then, $f$ is said to be *permutation equivariant* if for every input $\mathbf{x} = (x_1, x_2, ..., x_n) \in \mathbb{R}^n$, it holds that

$$f(\pi(\mathbf{x})) = \pi(f(\mathbf{x})), \tag{24}$$

where $\pi(\mathbf{x})$ represents the permutation of the components of $\mathbf{x}$ according to $\pi$, and $\pi(f(\mathbf{x}))$ represents the permutation of the components of $f(\mathbf{x})$ in the same manner. We can describe this property in a matrix form as follows: A function $F : \mathbb{R}^{n \times d} \to \mathbb{R}^{n \times d}$ is permutation equivariant if for any input matrix $\mathbf{X} \in \mathbb{R}^{n \times d}$ and any permutation matrix $P_\pi \in \mathbb{R}^{n \times n}$ (representing permutation $\pi$ of the $n$ items/rows), the following holds:

$$F(P_\pi \mathbf{X}) = P_\pi F(\mathbf{X}). \tag{25}$$

## F. Permutation in Multi-head Self Attention

### F.1. Multi-head Self Attention Structure

The MHA architecture proposed in (Vaswani et al., 2017) is a key component of the Transformer model. Let $n$ be the input (sequence) length, $d$ the model (embedding) dimension per token, and $h$ the number of parallel attention heads. We first project the inputs into query, key, and value matrices $\mathbf{Q}, \mathbf{K}, \mathbf{V} \in \mathbb{R}^{n \times d}$, where $\mathbf{Q} = \mathbf{X}\mathbf{W}_i^Q$, $\mathbf{K} = \mathbf{X}\mathbf{W}_i^K$, and $\mathbf{V} = \mathbf{X}\mathbf{W}_i^V$. Subsequently, we apply scaled dot-product attention independently in each of the $h$ heads (of width $d_k = d/h$), yielding head outputs $\mathbf{O}_1, \ldots, \mathbf{O}_h \in \mathbb{R}^{n \times d_k}$. Finally, we concatenate and linearly re-project:

$$\text{MHA}(\mathbf{Q}, \mathbf{K}, \mathbf{V}) = \text{Concat}(\mathbf{O}_1, \ldots, \mathbf{O}_h)\mathbf{W}^O, \tag{26}$$

where $\mathbf{W}^O \in \mathbb{R}^{(h\,d_k) \times d}$ is a trainable matrix. Each $i$-th head, also referred to as self-attention, is then defined as:

$$\mathbf{O}_i = \text{Attention}(\mathbf{Q}, \mathbf{K}, \mathbf{V}) \tag{27}$$

$$= \text{Attention}(\mathbf{X}\mathbf{W}_i^Q, \mathbf{X}\mathbf{W}_i^K, \mathbf{X}\mathbf{W}_i^V) \tag{28}$$

$$= \text{softmax}\left(\frac{\mathbf{X}\mathbf{W}_i^Q(\mathbf{X}\mathbf{W}_i^K)^\top}{\sqrt{d_k}}\right)\mathbf{X}\mathbf{W}_i^V \tag{29}$$

where $\mathbf{W}_i^Q, \mathbf{W}_i^K, \mathbf{W}_i^V \in \mathbb{R}^{d \times d_k}$ are trainable parameter matrices.

### F.2. Proof of Permutation Equivariance

**Proof.** Permutation equivariance means that if you permute the input, the output should be permuted in the same way. Mathematically, for a function $f$ and a permutation matrix $P$, this is expressed as:

$$f(PX) = P \cdot f(X) \tag{30}$$

Let $P\mathbf{Q}, P\mathbf{K}, P\mathbf{V}$ be the permuted version of the query, key, and value matrices $\mathbf{Q}, \mathbf{K}, \mathbf{V} \in \mathbb{R}^{n \times d}$:

$$P\mathbf{Q}, P\mathbf{K}, P\mathbf{V} \tag{31}$$

where $P$ is a permutation matrix. The attention mechanism with the permuted inputs can be described as:

$$\mathbf{O}'_i = \text{softmax}\left(\frac{P\mathbf{X}\mathbf{W}_i^Q\left(P\mathbf{X}\mathbf{W}_i^K\right)^\top}{\sqrt{d_k}}\right)P\mathbf{X}\mathbf{W}_i^V \tag{32}$$

$$= \text{softmax}\left(\frac{P\mathbf{X}\mathbf{W}_i^Q(\mathbf{W}_i^K)^\top\mathbf{X}^\top P^\top}{\sqrt{d_k}}\right)P\mathbf{X}\mathbf{W}_i^V \tag{33}$$

Using the property of the softmax function:

$$= P\cdot \text{softmax}\left(\frac{\mathbf{X}\mathbf{W}_i^Q(\mathbf{W}_i^K)^\top\mathbf{X}^\top}{\sqrt{d_k}}\right)P^T P\mathbf{X}\mathbf{W}_i^V \tag{34}$$

Since $P^\top P = I$ (the identity matrix, because $P$ is a permutation matrix):

$$= P\cdot \text{softmax}\left(\frac{\mathbf{X}\mathbf{W}_i^Q(\mathbf{W}_i^K)^\top\mathbf{X}^\top}{\sqrt{d_k}}\right)\mathbf{X}\mathbf{W}_i^V \tag{35}$$

$$= P\cdot \text{softmax}\left(\frac{\mathbf{X}\mathbf{W}_i^Q\left(\mathbf{X}\mathbf{W}_i^K\right)^\top}{\sqrt{d_k}}\right)\mathbf{X}\mathbf{W}_i^V \tag{36}$$

Hence:

$$\mathbf{O}'_i = P\mathbf{O}_i \tag{37}$$

Concatenating across all heads:

$$\text{Concat}(\mathbf{O}'_1, \ldots, \mathbf{O}'_h) = \text{Concat}(P\mathbf{O}_1, \ldots, P\mathbf{O}_h) \tag{38}$$

$$= P\cdot \text{Concat}(\mathbf{O}_1, \ldots, \mathbf{O}_h) \tag{39}$$

Finally, the equation can be arranged as:

$$\text{MHA}(P\mathbf{Q}, P\mathbf{K}, P\mathbf{V}) = P\cdot \text{MHA}(\mathbf{Q}, \mathbf{K}, \mathbf{V}) \tag{40}$$

This holds the definition of (30) and shows that the MHA mechanism is permutation equivariant, as the output under any permutation of the inputs is simply the same permutation applied to the original output. □

