# OpenReview forum: "Sortformer: A Novel Approach for Permutation-Resolved Speaker Supervision in Speech-to-Text Systems"
_ICML.cc/2025/Conference — ICML 2025 poster_

### Official Review · Reviewer_US9G · 2025-03-09

**Overall Recommendation:** 2

**Summary:**

This paper proposes a new multi-speaker speech diarization and recognition model, with a loss function bridging timestamps with the tokenzied texts, built on classification loss, permutation invariant loss, and the newly-proposed sort loss. The sort loss looks like an variant of PIL, where the label is placed in certain order, referred by either time or any relevant metric.

**Claims And Evidence:**

One important claim in the paper, or one of the main selling point of the paper, is that the SortFormer model cna reduce the workload of architectural adjustments of the original EEND model, and make the multi-speaker ASR training equivalent to mono-speaker one. However, this brings two problems:
1. The speaker supervision of the multi-speaker is still involved in the training, the sort loss brings it cleverly to the label side and use the loss function as a complmentary one of the permutation loss. So, either the claim itselfm or the novelty of the claim seems limited.
2. The paper does not show explicitly about how the model has been "minimally" adjusted.

**Essential References Not Discussed:**

The reviewer thinks there is no essential reference missed.

**Experimental Designs Or Analyses:**

The reviewer checks all the experimental results and setups.

1. The main problem is that under diarization results, sort loss works best when being complementary with the original PIT loss, and does not perform that well when using it alone, whose reason is not discussed in detail.

2. Besides, the training data for different models seems not well-unified and properly arranged. Since this is architectural and training strategy improvement, the authors should emphasize and be more regor on this issue.

**Methods And Evaluation Criteria:**

About the proposed method. The reviewer thinks think the novelty of the method is limited, especially when the sort loss
1. Still take advantage of the speaker supervision, and increases the computational load
2. Does not show significant advantage by itself.

The evaluation criteria in terms of datasets and metrics are good, following state-of-the-art approaches.

**Other Comments Or Suggestions:**

The reviewer does not have further comment or question about this paper, as such have been enlisted in detail in the earlier questions.

**Other Strengths And Weaknesses:**

There are several other minor weaknesses of the paper.
1. The significance of the method.
2. The efficiency of the method. The method itself involves token sorting accoridng to certain metric multiple times. The reviewer wonders if this will significantly increase the computational cost
3. Lacking clarity. There are multiple places in the paper lacking clarity or causes confusion. For example, the

**Questions For Authors:**

The questions for the authors of this paper has been enlisted in the above questions.

**Relation To Broader Scientific Literature:**

The key contribution of the paper is about the proposed multi-class loss function, which takes good advantage of the speaker labels and timestamps. However, this may limit the contribution of thenpaper to multi-speaker ASR where there is no overlap between the speakers (and preferrably, long pauses between them). This is not shown in the demo (supplementary material) needs to be validated.

**Theoretical Claims:**

There are several theoretical claims in the appendix F.2. about permutation of multi-head self attention (MHA):
1. MHA is permutation invariant
2. MHA is permutation equivariant

However, the 1st claim's proof seems not correct, especially 29) to 30) needs more detailed explanation.

---

> ### Author Rebuttal · Authors · 2025-04-01
>
> ## Response to Claims and Evidence
>
> 1. Sort loss is not a complementary or supportive loss to the system. Only by including Sort Loss, the model learns to arrange speaker predictions in arrival-time order.
>
>    a. If we only use PIL, the model does not have arrival-time sorting capability.
>    b. We clearly state the goal of Sort Loss in the abstract: It is designed to bridge the gap between ASR tokens and diarization timestamps during “multispeaker ASR training”. And this is achieved by training the diarization model using Sort Loss or Sort+PIL hybrid loss.
>    c. Therefore, it is fair to say rather PIL (in the hybrid loss) is playing a role as a supplementary loss since it additionally corrects the mapping between predictions and labels since sort-prediction could be erroneous in some cases.
>
> 2. The “minimal architectural adjustments” are clearly explained throughout the paper.
>
>    a. Section 2.3, Section 3.1 and Figure. 2 Explain the downside of modular or pipelined multi-speaker ASR systems and we explain how our proposed method can train a multi-speaker ASR model without applying any specialized loss but using the standard token-level cross-entropy loss.
>    b. If you want to visually check the “minimal architectural adjustments”, see “Sortformer” module and sinusoidal kernels in Figure 2. The original modules in the vanilla Carany model (FastConformer and Transformer Decoder) are not altered in terms of architecture. We optionally use adapters to boost the performance with less amount of trainable weights.
>    c. In summary, our proposed approach maintains the original ASR model’s architecture, except the part where we inject encoder output with speaker-kernel injection.
>
> ## Response to Theoretical Claim
>
> 1. We found couple of errors in the appendix F:
>
>    a. In F2 and F3, all QW^{Q}, KW^{K}, VW^{W} should be changed to XW^{Q}, XW^{K}, XW^{W}.
>    b. Regardless of the mistake a, we did not put a necessary condition “Q is learned parameters” to permutation invariance (See-[https://people.tamu.edu/~sji/classes/attn-slides.pdf]). Thus, the proof F.2 is incorrect and we will remove the proof of permutation invariance (F.2) in the final version. We do not use learnable query Q in our proposed system either.
>    c. The Permutation Equivariance property (when there is no positional embedding) seems enough for supporting the need for positional embedding.
>
> ## Response to Relation To Broader Scientific Literature
>
> We believe that the reviewer’s claim “this may limit the contribution of the paper to multi-speaker ASR where there is no overlap between the speakers (and preferably, long pauses between them)” can hardly be considered a valid criticism, because:
>
> 1. The supplementary demo video clip is designed for reviewers to intuitively understand multi-speaker ASR tasks in action, not to assess the functionality and performance of the proposed system.
>
> 2. There is plenty of evidence that our proposed system works without such issue:
>
>    a. More than half of the samples in the diarization datasets DIHARD3, NIST-SRE-2000, CH109, are longer than 5 minutes, and these datasets have a very frequent long gap between each speaker’s segment. In the multi-speaker ASR test set AMI Corpus, the overlap ratio is 14% and the rest of 86% speech is non-overlapping speech.
>    b. Based on these reasons, the point raised by the reviewer’s claim “this may limit the contribution” doesn't seem like a reasonable concern.
>
> ## Response to Other Strengths And Weaknesses
>
> 2. The additional computational cost could be discussed as follows:
>
>    a. Training Time Impact vs pure PIL training
>    ∙ Pure Sort Loss: +0.23% (17.038 vs 17.0 min/epoch)
>    ∙ Hybrid Loss: +2.26% (17.385 vs 17.0 min/epoch)
>
>    b. Inference Time of Multispeaker ASR: LibriSpeechMix test-3mix (2,620 files, total duration: 42,514.9s) on an NVIDIA RTX A6000, batch size 100, with 10-run averages.
>    - MS-ASR with Sortformer: 300.213s
>    - MS-ASR without Sortformer 297.891s
>    - Adding Sortformer causes 0.78% overhead in runtime (1.0078x)
>
>    c. Inference on standalone diarization: No added inference time compared to PIL trained models.
>
>    d. Our view on additional computational cost:
>    (1) Training Sortformer with Hybrid Loss (Sort Loss + PIL) increases runtime by only 2.26% (x1.0226) compared to PIL-only training, while Sort Loss alone adds just 0.23% (x1.0023).
>    (2) For inference, standalone diarization (Sortformer) adds no extra time versus the PIL-trained model.
>    (3) Multispeaker ASR inference time increases marginally by 0.78% (x1.0078) when using Sortformer.
>    (4) On the LibriSpeechMix dataset, our system achieves a 25.6% relative error rate reduction (7.14% → 5.31%).
>    (5) Given the significant performance gains, the added computational cost is quite negligible, making the claim of "significant computational cost" not very convincing.
>
> 3. The reviewer’s comment seems unfinished: “For example, the”.

---

> > ### Comment · Reviewer_US9G · 2025-04-06
> >
> > Thanks the feedback from the authors and the reviewer would like to apologize about the unfinished sentence. Please ignore that part.
> >
> > However, the reviewer cannot agree with multiple points in the rebuttal of the authors, such as:
> > 1. "The supplementary demo video clip is designed for reviewers to intuitively understand multi-speaker ASR tasks in action" - I think the reviewers of this paper shall normally have knowledge about multi-speaker ASR, so a lecture demo video is not necessary. Besides, the purpose of this video is not clarified in the paper, nor in the supplementary material. If the reviewer has overlooked anything, would like to learn more.
> > 2. The authors' response to the experimental novelty is not sufficient. After show-casing the numbers, it does not answer the critique about the model "Does not show significant advantage by itself".
> >
> > The reviewer hopes the authors can address these concerns.

---

> > > ### Author Response · Authors · 2025-04-07
> > >
> > > ## Rebuttal on the weakness and significance of the method.
> > >
> > > ### (1) The original review's points are well refuted.
> > >
> > > #### 1. Computational Load
> > > - We showed the numbers on both training and inference in the first rebuttal comments.
> > >
> > > - a. Training: Hybrid Loss increases +2.23% training time.
> > > - b. Inference: A multispeaker ASR model with Sortformer has an 1.0078x times increased runtime compared to a multispeaker ASR model without Sortformer.
> > >
> > > - This is very hard to be regarded as a significant increase in computational load.
> > >
> > > #### 2. Significance in benefit
> > >
> > > - We stated that "On the LibriSpeechMix dataset, our system achieves a 25.6% relative error rate reduction (7.14% → 5.31%)."
> > > - This result is also outperforming all the other previous studies that reported all three types of mixtures on LibriSpeechMix.
> > > - The reviewer's view on the significance of the method can be always subjective. However, the reviewer's claim on "lacking significance" does not have any supporting arguments.
> > > - For example, the reviewer can mention previous studies that have similar concepts and performance, lack of diversity in the evaluation datasets or practical limitations that will arise in the real-life scenario. None of these reasons were claimed except claiming "This work has a limited significance".
> > >
> > > ### (2) The weakness claims are repeatedly mentioned in the original review in multiple sections without any changes.
> > >
> > > #### **Repeated Weakness Claims 1**:
> > > - In the "Methods And Evaluation Criteria", the reviewer says - "Does not show significant advantage by itself."
> > > - In the "Other Strengths And Weaknesses", the reviewer repeats - "The significance of the method." without mentioning any new points.
> > >
> > > #### **Repeated Weakness Claims 2**:
> > > - In the first review, there was a comment "Still take advantage of the speaker supervision, and increase the computational load".
> > > - The same type of comment is repeated in "Other Strengths and Weaknesses" saying "The efficiency of the method.", without any new points.
> > >
> > > ### (3) There are technically wrong review comments
> > >
> > > #### Technically wrong descriptions 1:
> > >
> > > The reviewer's comment - "The key contribution of the paper is about the proposed multi-class loss function, which takes good advantage of the speaker labels and timestamps." is not technically accurate, because:
> > >
> > > - There are no "taken advantages" from the speaker labels and timestamps. Our method takes advantage of the "Arrival time sorting" mechanism from the transcription and diarization model. This is a completely inaccurate description of what is being done.
> > >
> > > #### Technically wrong descriptions 2:
> > >
> > > The reviewer's comment - "This may limit the contribution of thenpaper to multi-speaker ASR where there is no overlap between the speakers (and preferrably, long pauses between them)."
> > >
> > > - This is technically wrong comments because: the evaluation datasets we used (NIST SRE 2000, CH109) include samples with lots of silence between the speaker's speech. And also there are many sessions in DIHARD3, CH109 where there are no overlaps at all. Our system shows equally good performance on these datasets. There is no evidence that it would not work or limited benefits will be made on such samples.

---

### Official Review · Reviewer_98zA · 2025-03-10

**Overall Recommendation:** 3

**Summary:**

The authors introduce Sortformer, a model built on a transformer-based encoder and trained using a hybrid loss that combines permutation invariant loss (PIL) and the newly proposed Sort Loss. Sort Loss is formulated as a binary cross-entropy loss, calculated between the sorted speaker presence labels in a sequence and the encoder’s output sequence. Once trained, the model can be seamlessly integrated with a multi-speaker ASR system, which is trained using standard cross-entropy loss. The method was evaluation on diarization and multi-speaker ASR tasks.

**Claims And Evidence:**

* The claim that Sort Loss alone solves the permutation problem independently of PIL is not fully supported by the results. The experiments indicate that Sort Loss can complement PIL but cannot replace it. This is evident from the use of the weight parameter $\alpha$ = 0.5, which gives equal importance to the PIL term.

* The proposed model is designed for both speaker diarization and multi-speaker ASR. It has been evaluated on relevant benchmarks, and the results show that while it helps reduce word error rate (WER) in multi-speaker ASR, its impact on improving diarization error rate (DER) is less significant.

* A deeper analysis of the proposed loss function is needed. Specifically, it would be helpful to examine how it affects training time and why its performance degrades as the number of speakers increases.

**Essential References Not Discussed:**

The essential references are discussed.

**Experimental Designs Or Analyses:**

The design and evaluation looks reasonable to me.
There are missing analysis sections: $\alpha$ parameter tuning, time measurements, robustness to noise, error analysis.

**Methods And Evaluation Criteria:**

The method was evaluation on diarization and multispeaker asr tasks.
On diarization taks - the authors used 3 test sets and compared to six recent baselines.
On multispeaker ASR - the model was tested on AMI test and CH109 test set and compared to baseline on LibriSpeechMix.
Evaluation looks reasonable to me.

**Other Comments Or Suggestions:**

* Figure 4 and Table 1 - need a space margin at the bottom
* Equation 14 - not clear what is $A$

**Other Strengths And Weaknesses:**

**Strengths**:
* The paper is well-written with clear explanations and visualizations.
* The proposed method shows strong performance on multi speaker ASR task (Table 3)


**Weaknesses:**
* According to Table 1 the proposed model works well for 2-3 speakers and the performance degrades as for 4 speakers.
* Compared to PIL - the proposed loss doesn’t propose any improvements for diarization task (Table 1 - *Sortformer-PIL* and *Sortformer-Sort-Loss* rows)
* Limited Robustness Analysis: Needs more tests on low-resource languages and extreme noise conditions.

**Questions For Authors:**

* What is the loss used for Multi-speaker ASR Training Data? Please describe it in the paper.

* Are there any limitations of the proposed model? Please add a section in the text or in the appendix.

**Relation To Broader Scientific Literature:**

* Sortformer is one of the first models to integrate speaker diarization directly into ASR models using a differentiable sorting-based loss.

*  The Sorted Serialized Transcript (SST) approach simplifies multi-speaker ASR training to be similar to mono-speaker ASR training.

**Theoretical Claims:**

There are no theoretical claims.

---

> ### Author Rebuttal · Authors · 2025-04-01
>
> ## Response to Claims And Evidence
>
> ### 1. Response to the reviewer's claim that the statement "Sort Loss solves the permutation problem is not fully supported":
>
> a. What we cannot achieve without PIL+ alpha=0.5 is the "maximized diarization performance", not "resolved permutation".
> b. Therefore, we believe we can say that "Sort Loss based training" resolves the permutation problem in speaker diarization training and multispeaker ASR training.
> c. Even if we use only Sort Loss without PIL, the Sortformer still performs with the sorting capability and PIL is a helping hand.
>
> ### 2. WER/DER with Sort Loss
> Showing the SOTA diarization with Sort Loss is not the contribution we claim.
>
> ### 3. Response to missing deeper analysis:
>
> a. Given page constraints, we focused on demonstrating how Sort Loss enables diarization supervision using standard ASR token-level cross-entropy loss.
>
> b. Table 1 shows all comparable systems exhibit similar DER degradation with more speakers, as each additional speaker introduces compounding diarization errors with the similar proportion as our proposed system. The sole exception (WavLM-EEND-VC) is not an end-to-end speaker diarization model, weakening the validity of direct comparisons.
>
>
>
>
> ## Response to Experimental Designs Or Analyses
>
> ### 1. Alpha parameter tuning:
> We did grid search on this alpha parameter then found that 0.5 is the best performing parameter. We will add the grid-search results in the appendix in the final version.
>
> ### 2. Runtime measurements:
>
> #### a. Training Time Impact vs pure PIL training
>
> - Pure Sort Loss: +0.23% (17.038 vs 17.0 min/epoch)
> - Hybrid Loss: +2.26% (17.385 vs 17.0 min/epoch)
>
> #### b. Inference Time of Multispeaker ASR:
> LibriSpeechMix test-3mix (2,620 files, total duration: 42,514.9s) on an NVIDIA RTX A6000, batch size 100, with 10-run averages.
>
> - MS-ASR with Sortformer: 300.213s
> - MS-ASR without Sortformer 297.891s
> - Adding Sortformer causes 0.78% overhead in runtime (1.0078x)
>
> #### c. Inference on standalone diarization:
> No added inference time.
>
>
> ## Response to Other Strengths And Weaknesses
>
> ### 1. In the Table 1, we wanted to show:
>
> a. Showing Sort Loss achieves SOTA is not the key contribution.
> b. Sort-Loss can perform at a comparable level with PIL-trained models.
> c. Hybrid Loss outperforms PIL based method, without losing "learned arrival-time sorting capability".
>
> ### 2. Sort Loss shows no improvement over PIL
> We believe this is a minor weakness because we clearly state the goal of Sort Loss in the paper. Resolve permutation during multispeaker ASR training bridging diarization timestamps and tokens. Improving the PIL method with Sort Loss is not the main contribution we claim.
>
> ### 3-1. Robustness:
> One of our evaluation dataset, DIHARD3, as the name suggests "Diarization is Hard", includes 11 domains that are very challenging to diarize such as noisy restaurant conversations, web-video and street interviews. DIHARD3 is the most noisy and reverberant diarization evaluation dataset.
>
> ### 3-2. Low-resource language:
> NIST SRE 2000 dataset includes multi-lingual speech (Mandarin Chinese, Vietnamese, Spanish, Tagalog, etc) and DIHARD3 also includes Arabic, Mandarin, Min Nan, Portuguese, Russian, and Polish language. Therefore, the evaluation on these datasets involve diarization performance on low-resource languages.
>
> ### 3-3. Missing evaluations
> Hence, we do not think that the Sortformer diarization model is completely missing evaluations on noise robustness and low-resource languages.
>
> ## Response to Other Comments Or Suggestions
>
> 1. We realized this margin crash only after the submission. We will fix this in the final version of the paper.
> 2. "A" in Equation (14) is the output state (also referred to as ASR embedding) from Fast Conformer encoder in Figure 2. We will specify this in the final version.
>
> ## Response to Questions For Authors
>
> ### 1. “What is the loss used for Multi-speaker ASR Training Data?”
> The answer to this question is “Cross Entropy Loss” and it is already clearly mentioned multiple times in the manuscript:
>
> a. The last paragraph in Section 1 (Introduction) “multi-speaker ASR training ... tokenlevel cross-entropy loss”
> b. Figure 2. See “Cross-Entropy Loss” at the top with the loss function symbol “L_{CE}”.
> c. End of Section 2.3 “Our approach focuses ... based on token objectives and cross-entropy loss”
> d. End of Section 3.3 “the model can be trained ... cross-entropy function..”
> e. In Section 6 (Conclusion), “... thereby supporting cross-entropy loss-based training and unifying the multi-speaker ASR framework...”
>
> ### 2. Limitations of our proposed system:
>
> a. Currently, our proposed system has a limited inference length of 45 seconds and limited maximum of 4 speakers in a session.
> b. The implementation we use for the experimental results runs Sortformer and FastConformer encoder in serial manner. This adds up inference time to the total inference time although it is less than 1%.

---

> > ### Comment · Reviewer_98zA · 2025-04-04
> >
> > Thank you for the detailed response.
> > * I think the representation and explanation of  the losses that are used at different training setups should be improved. I see that ce loss is used for ASR training but I'm not sure how the Sortformer is optimized when it is integrated with ASR, do you train it before or jointly with ASR? Where can I see this in the manuscript?
> >
> > * When you say that the main contribution is *Resolve permutation during multispeaker ASR training bridging diarization timestamps and tokens*, what results are reflecting this contribution in the evaluation section?

---

> > > ### Author Response · Authors · 2025-04-07
> > >
> > > ---
> > >
> > > ## A. Comment on description of losses used for each phase of training.
> > >
> > > ### (1) Acknowledgement of the lack of concise summaries
> > >
> > > - We admit that the descriptions of training or fine-tuning Sortformer could be clearer and more prominently emphasized, especially for readers who are not very familiar with the field and multi-speaker ASR training tasks.
> > >
> > > - In the abstract or introduction, there are no sentences that concisely explain the specific aspects the reviewer mentioned.
> > >
> > > ### (2) Our plans for revision
> > >
> > > We will definitely add a concise and clear summary of the parts you identified as unclear, if our paper is accepted for publication at ICML 2025.
> > >
> > > ### (3) The parts of the manuscript that mentions the reviewer questioned
> > > Regarding the question: "Do you train it before or jointly with ASR? Where can I see this in the manuscript?" — we believe this point was explained fairly clearly in the manuscript as follows:
> > >
> > > - In Section 4.3, See the line that says: "Thus, without using the PIT or PIL approach, we can calculate the loss from the speaker tokens to train or fine-tune both the Sortformer diarization model and ASR model"
> > > - In Section 5.3, See the line that says: "System 2 and System 3 are the models where Sortformer diarization module is plugged in while Sortformer model weights are frozen in System 2 and fine-tuned in System 3."
> > > - In Table 2., See the column named "Diar Model Fine-tune". System 2,5 and 6 are cross marked because the Sortformer model was frozen, while System 3 is fine-tuning the Sortformer model so it has a "check mark".
> > > - Since the above explanations can be only understood after reading the manuscript, we notified the Sortformer box in Figure 2 with "fire" icon and "frozen" icon, which means this module can be either fine-turned or frozen during the process. (We believe these icons represent a universally understood meaning in the machine learning field.)
> > > - In Section 5.3.2, we mentioned Table 3 results are based on frozen Sortformer training by saying: "Then we run 180K steps of fine-tuning of the ASR model while keeping the Sortformer model frozen,"
> > >
> > > ### (4) Our rebuttal on the reviewer's comment
> > >
> > > - We admit that whether the Sortformer is fine-tuned during multi-speaker ASR training is neither emphasized nor summarized in the introduction or abstract.
> > >
> > > - We did not emphasize or reiterate the fine-tuning of Sortformer during multi-speaker ASR training because it did not significantly improve performance across all datasets.
> > >
> > > - That said, the technical description of whether Sortformer is fine-tuned is clearly presented in the manuscript. This should not be viewed as an omission of experimental conditions or a lack of technical detail.
> > >
> > > - **If the reviewer is reconsidering the review outcome score, we hope they will take our position into account: we acknowledge and accept that this point was not sufficiently emphasized and must be added — however, we emphasize that the relevant details are already described in the manuscript.**
> > >
> > > ---
> > >
> > > ## B. Comment on "What results are reflecting this contribution in the evaluation section?"
> > >
> > > The short answer is, it is in Table 2 and Table 3.
> > >
> > > ### (1) Acknowledgement of the limited clarity of the description
> > >
> > > - We admit that we did not explicitly use the phrase "resolving the permutation" in Section 4.3.
> > >
> > > - However, this was intentional to minimize confusion, as we are using the standard token-level cross-entropy loss on the sorted serialized transcript (e.g., "<|spk0|> hi how are you, <|spk1|> good you · · <|spkK|>"). We do not refer to this with any specialized terminology.
> > >
> > > - We expected the readers to figure matching the speakers in the transcription "<|spk0|> <|spk1|> <|spkK|>" with the speaker bin 0,1,2 and 3 in Sortformer output's speaker dimension is "permutation resolving".
> > >
> > > ### (2) Our plans on revision
> > > We will definitely add a clear description stating, "This is how we resolve permutation in multi-speaker ASR training," if our paper is accepted for publication at ICML 2025.
> > >
> > > ### (3) Our rebuttal on the reviewer's comment
> > >
> > > - We also acknowledge that the exact phrase "resolving permutation" is not used in Sections 4 or 5. Table 2 and Table 3 present results obtained using our proposed method, which involves our permutation-resolving technique.
> > >
> > > - We explained in the introduction and related work that no prior studies have used this type of permutation-resolving technique to train an end-to-end multi-speaker ASR system.
> > >
> > > - Accordingly, we expected readers perceive the whole process of using Sortformer, arrival time sorted speaker supervisions from Sortformer and sorted serialized transcription as "resolving permutation".
> > >
> > > - **If the reviewer is willing to change the review outcome score, we hope the reviewer can take our take on the "resolving permutation" phrase.**
> > >
> > > ---
> > > ## C.
> > > Please consider we have added run-time analysis on training and inference. Our proposed method adds very minimal amount of training time and inference time.

---

### Official Review · Reviewer_rtb3 · 2025-03-12

**Overall Recommendation:** 4

**Summary:**

This paper proposes a model called sortformer and a sorting loss to achieve joint speaker diarization and ASR without the need for permutation invariance loss. The proposed model can still be trained with PIL and it can also be combined with the sorting loss. In terms of modeling, the speaker label probabilities are obtained by setting up the output layer as a multilabel output layer with sigmoids. To get the sorting loss, speaker are labeled based on their appearance order in time. Hence the first speaker is labeled as class 0, the next one as class 1, etc. Once the ground truth speakers are also sorted in a similar way, the sorting loss uses binary CE. Speaker supervision to the model is provided by the kernel based speaker encodings. Training of the joint ASR + diarization model can be done either at word or segment level.

Experiments use real and simulated speech mixtures for training and tested on various test sets in terms of DER for speaker diarization and WER for ASR. The results presented suggest competitive performance as compared to the existing approaches with a simpler training framework.

## update after rebuttal
I would like to keep my score after the rebuttal.

**Claims And Evidence:**

Seems correct.

**Essential References Not Discussed:**

NA

**Experimental Designs Or Analyses:**

In table 1, different loss functions and the use of post-processing are compared, resulting in hybrid loss with post-processing leading to the best performance (expected). In 2- and 3-speaker conditions, Sortformer outperforms existing approaches. For n=4 speakers, the performance is slightly behind than a previous study but still the numbers are comparable.

From Table 2, it seems that the bigger model (model 6 vs. 2) performs better. Hence, it would be good to know the sizes of the models in Table which outperformed the Sortformer model on some test sets. Could you please add those details to Table 1?

ASR experiments on LibrispeechMix also show promising WERs.

**Methods And Evaluation Criteria:**

Yes

**Other Comments Or Suggestions:**

- Fig. 5. might be updated in light of the question I raised above.

**Other Strengths And Weaknesses:**

+ Strengths: The paper proposes a simple but working solution to permutation invariance issues in speaker diarization. The proposed loss could be used by itself or in combination with PIL which can be useful for other model training applications.

- Weakness: A few additional experimental experiments could have made the paper more stronger. For example, what if we replaced the sinusoidal guidance with some sort of real speaker embeddings from a speaker recognition system?

- In terms of clarity, the paper is clear overall. However, a few details might have been made clearer. For example, Fig. 5, shows a scenario where the green speaker speaks twice with a different speaker (spk4) in between those two segments. In the proposed system, how do we treat the second section of the green speaker? Do we still say spk3 or do we increment the count and say spk5?

**Questions For Authors:**

1) What if we replaced the sinusoidal guidance with some sort of real speaker embeddings from a speaker recognition system, how would the diarization and ASR results look like?

2) Fig. 5, shows a scenario where the green speaker speaks twice with a different speaker (spk4) in between those two segments. In the proposed system, how do we treat the second section of the green speaker? Do we still say spk3 or do we increment the count and say spk5?

3) In Section 5.4, it is mentioned that "System 5 not only shows degradation in segment-level objectives," Do you have an explanation for this result which you can add to the text?

**Relation To Broader Scientific Literature:**

Sortformer, and especially the sorting loss can be used as a complementary tool for applications that require permutation invariant training. The most common use case is the speaker diarization domain but the idea can be utilized in other type of applications, too.
Use of sinusoidal embeddings for speaker guidance is also an interesting idea.

**Theoretical Claims:**

Read the equations quickly. They seem to be correct.

---

> ### Author Rebuttal · Authors · 2025-04-01
>
> ## Response to Experimental Designs Or Analyses:
>
> Here is model size information for the models we listed in Table 1. We will add the model size to Table 1.
>
> - **MSDD**: 31.1 M
> - **EEND-EDA**: 6.4M
> - **WavLM-L + EEND-VC**: 317M+
> - **EEND-GLA-Large**: 10.7M
> - **AED-EEND**: 11.6M
> - **AED-EEND-EE**: 11.6M
>
> ## Response to Other Strengths And Weaknesses:
>
> 2-1. Comments on speaker embedding experiments: We have done a few experiments by concatenating speaker embedding (TitaNet) for the same multispeaker ASR task. However, we were not able to obtain any improvements with both frozen and fine-tunable speaker embedding extractors. The reason we are making a guess is as follows:
>
> a. The speaker embedding model is trained to distinguish 10K to 100K different speakers and the speaker representations are too complicated to make synergistic effect on 2~4 speakers in a session.
>
> b. Not only speaker tagging accuracies, speaker embedding interferes with the ASR output states (ASR embedding from Fast Conformer Encoder) and damages the WER itself.
>
> c. Since there is already plenty of content included in the paper, we did not include the speaker embedding experiments which have no improvements. In addition, concatenating speaker embedding for speaker adaptation or multi-talker ASR is not a novel idea, which has been already tried in many of previous studies. We did not think that showing speaker-embedding concatenation experiments bolster the main idea of this paper and the other contents should be prioritized to explain and demonstrate the proposed method.
>
> 2-2. Speaking of the speaker embedding model, Sortformer is initialized with the NEST (Nemo Encoder for Speaker Tasks) model that is self-supervised learned on large size unlabeled dataset and NEST can be trained to perform speaker verification/identification. In a way, we are already taking advantage of speech representation pretrained on large amounts of datasets.
>
> 3. The second section of the green speaker should still be spk3. Not only our system but this applies to all the speaker diarization systems or multi-speaker ASR. If you want to have a better understanding on multi-speaker ASR tasks, please watch the supplementary demo video that contains multi-speaker ASR in action.
>
> ## Response to Other Comments Or Suggestions:
>
> - We will update Figure 5 to intuitively explain the issues you highlighted in the final version.
>
> ## Response to Questions For Authors:
>
> 1. Please see rebuttal for “Other Strengths And Weaknesses”.
> 2. Please see rebuttal for “Other Strengths And Weaknesses”.
> 3. We speculate that this happens because word-level objectives produce more gradient for speaker tagging than segment level objectives due to its token count.
>
> a. The segment-level objective includes a speaker token whenever there is a speaker switch (i.e., speaker change), whereas in the word-level objective, each word is accompanied by a speaker token.
>
> b. In the proposed MS-Canary system, we compute the cross-entropy (CE) loss on the multispeaker text output (consisting of both speaker and text tokens) for model training.
>
> c. Consequently, the word-level objective places greater importance on correct speaker assignment in addition to accurate word recognition compared to the segment-level objective. This leads to an improved cpWER.
>
> ## Side Note
>
> Please also take a look at the other rebuttals for other reviewers regarding runtime measurements.

---

### Official Review · Reviewer_cJRr · 2025-03-13

**Overall Recommendation:** 3

**Summary:**

The paper proposes Sortformer, an encoder-based neural model designed for permutation-resolved speaker diarization integrated into speech-to-text (STT) systems. Its core innovation is the introduction of a Sort Loss that addresses the traditional permutation invariance problem in speaker diarization by sorting speakers based on their arrival time, rather than solely relying on permutation invariant loss (PIL). Additionally, the paper introduces a novel multi-speaker speech-to-text architecture which embeds sorted speaker labels using sinusoidal kernel functions directly into the encoder's hidden states. Experimental results demonstrate that using Sort Loss, especially in combination with traditional permutation invariant loss (PIL), boosts diarization and multi-speaker transcription accuracy. The authors claim that this framework simplifies the training of multi-speaker ASR to be as straightforward as mono-speaker ASR, facilitating integration into multimodal large language models (LLMs).

**Claims And Evidence:**

- The main claim—that Sort Loss resolves the speaker permutation ambiguity effectively—is well-supported through multiple experiments across standard diarization benchmarks (DIHARD3, CALLHOME, CH109, and LibriSpeechMix). The performance improvement when using hybrid loss (Sort Loss + PIL) is clearly demonstrated.
- The paper clearly states and experimentally validates that the proposed system simplifies the integration with existing speech-to-text architectures, as indicated by improved results when combined with models like Canary-170M and Canary-1B.
- Despite thorough experimental evidence, the paper does not clearly address the robustness of Sort Loss under highly complex, realistic scenarios involving noisy, reverberant, or strongly overlapping speech. Thus, claims about robustness and generalizability may be overstated or insufficiently explored.
- It is not fully demonstrated whether improvements in diarization accuracy directly translate to significantly better downstream NLU tasks, such as conversation summarization or information extraction, which are key use cases motivating the research.

**Essential References Not Discussed:**

The authors have generally cited essential works thoroughly.

**Experimental Designs Or Analyses:**

- The experimental design is largely sound, particularly the choice of datasets. Nevertheless, there are weaknesses: the authors use an artificially created LibriSpeechMix dataset with fixed delays, which is less representative of real-world spontaneous conversational scenarios. Evaluating the model on more realistic, noisy, or spontaneous speech scenarios (such as more diverse conversational datasets) would provide stronger evidence of real-world applicability.
- The training setup lacks extensive details about hyperparameter selection (particularly the choice of α in hybrid loss, the impact of dropout rates, and the reasons for not employing data augmentation techniques like SpecAugment), leaving unanswered questions regarding the robustness of the presented results.

**Methods And Evaluation Criteria:**

- The methods (Sort Loss, hybrid loss, and sinusoidal kernels) are well justified and designed specifically to address the permutation problem in speaker diarization. The paper thoroughly uses established datasets (CALLHOME, DIHARD3, LibriSpeechMix, AMI, ICSI) that reflect realistic and diverse scenarios.
- However, the proposed evaluation criteria, while sensible, could be enhanced by additional metrics assessing latency, resource efficiency, and robustness to domain shifts, which are critical in real-world scenarios but currently missing from evaluation criteria.

**Other Comments Or Suggestions:**

Please refer to the previous sections.

**Other Strengths And Weaknesses:**

**Strengths**
- The introduction of Sort Loss and integration into ASR frameworks is a significant conceptual innovation.
- Rigorous empirical results thoroughly demonstrate effectiveness and robustness.
Practical Significance: Easily integrable and adaptable to various downstream speech applications and multimodal LLM systems, increasing practical relevance.

**Weaknesses**
- Computational overhead or runtime comparisons were not thoroughly addressed, leaving open questions regarding practical efficiency.
- Sort Loss inherently assumes accurate estimation of arrival times. Error analysis of mis-sorting scenarios is somewhat limited, raising potential robustness concerns under challenging real-world conditions.

**Questions For Authors:**

- Can you provide detailed computational runtime benchmarks for Sortformer integration into ASR, especially compared to standard diarization modules? Understanding practical deployment feasibility is critical.
- How robust is Sortformer to initial sorting inaccuracies? Have you conducted experiments analyzing the system's degradation under varying levels of sorting errors, and if so, what were your findings?
- Could you clarify explicitly how Sort Loss compares conceptually and empirically to recent dominance-based methods like DOM-SOT [1]? Would such methods complement or potentially outperform Sort Loss under certain conditions?

[1] Shi, Ying, et al. "Serialized Output Training by Learned Dominance." Proc. Interspeech 2024. 2024.

**Relation To Broader Scientific Literature:**

Sortformer effectively addresses existing gaps in multi-speaker ASR and speaker diarization literature:
- It explicitly discusses the limitations of PIL-based diarization methods (EEND series) regarding integration complexity into ASR systems.
- It effectively positions itself within recent literature, notably SOT (Serialized Output Training) methods, multi-task learning in speech models, and transformer-based diarization architectures.
- Clear reference to recent competitive models (EEND variants, WavLM, DOM-SOT) is well contextualized.

The paper is notably strong in placing its novel Sort Loss clearly relative to prior work.

**Theoretical Claims:**

The paper provides detailed theoretical justifications related to permutation invariance and equivariance properties of multi-head attention mechanisms in Transformers (Appendices E and F). The proofs for these properties are checked thoroughly and appear mathematically sound and correct.

---

> ### Author Rebuttal · Authors · 2025-04-01
>
> ## 1. Response to Claims and Evidence
>
> ### 1-1. Robustness
>
> While the reviewer may find our evaluation insufficient, we believe the robustness of Sort Loss is demonstrated through the datasets we used up to a certain degree:
>
> - **DIHARD3** ("Diarization is Hard") covers 11 challenging domains, including noisy and reverberant scenarios (e.g., restaurants, street interviews, web videos). To our knowledge, no other benchmark is more comprehensive for noisy and reverberant speaker diarization.
>
> - **LibriSpeechMix** is specifically designed for overlapping speech evaluation, with overlap rates ranging up to 90%, including three-speaker overlaps. It is widely used in multi-talker ASR research.
>
> - We selected widely accepted benchmark datasets to ensure reliable comparisons. Evaluating on untested datasets would compromise result validity. We encourage the reviewer to reconsider after reviewing the datasets' features.
>
> ### 1-2. Comment on NLU Task Evaluation
>
> We do not think that evaluating multi-speaker ASR on NLU tasks is a critical omission:
>
> - In our target applications (e.g., meeting transcriptions, patient-doctor dialogues, and real-time speaker-tagged transcriptions), the primary output (text with speaker labels) is often consumed directly. Thus, metrics like DER and cpWER are more relevant.
>
> - Designing such an evaluation would require extensive methodology descriptions, which is beyond the scope of this 8-page paper. While NLU-based assessment is an interesting direction, we believe it warrants a separate study.
>
>
>
>
> ## 2. Response to Experimental Designs Or Analyses
>
> > "..LibriSpeechMix .. with fixed delays":
>
> The reviewer’s concern about “fixed delays” is incorrect. LibriSpeechMix is artificially mixed, but each session uses **randomized delays**. No fixed delays are applied.
>
>
>
>
>
> ## 3. Response to Questions For Authors
>
> ### 3-1. Runtime Evaluations
>
> Inference time on stand-alone diarization model:
>
> Dataset: LibriSpeechMix test-3mix (2,620 files, total duration: 42,514.9s) on an NVIDIA RTX A6000, with 10-run averages.
>
> Stand-alone diarization (batch size=1, collar=0.0, overlap included eval):
>
> - Pyannote-Diarization-3.1 (most popular open-source diarization model):
>   4m39.6s (RTFx=152.06), DER=0.2144, Speaker Counting Accuracy=0.4985
>
> - Proposed Sortformer (123M params):
>   2m14.6s (RTFx=316.01), DER=0.1346, Speaker Counting Accuracy=0.9763
>
> Multi-speaker ASR (batch size=100):
>
> - MS-Canary (170M params): 297.891s
> - Sortformer-MS-Canary (293M params): 300.213s
>   → Only 0.78% runtime increase (x1.0078) from adding Sortformer supervision.
>
> ### 3-2. Sorting Inaccuracies and Its Effect on Performance
>
> The model does not make sorting errors; the model only makes errors in diarization:
>
> - The Sortformer model does not make "sorting errors". We have never seen an example where Sortformer generates speaker segments’ arrival time in the wrong order. If trained with enough data and time, assigning speaker index in arrival time order is an easy task for the model.
>
> - The error comes from “missed” or “false alarm” predictions—i.e., short segments missed or non-existing segments falsely detected. This type of error occurs in all diarization systems.
>
> - Therefore, the starting time of the first speech segments of each speaker (t1, t2, t3, t4) are always in order: t1 < t2 < t3 < t4, even if the diarization is wrong.
>
> - For this reason, we are not able to perform “sorting error-based analysis.” We can only evaluate diarization errors and cpWER.
>
> ### 3-3. Scenarios Where Sorting-Based Mapping Could Be Inaccurate
>
> #### a. Training Phase
>
> - Table 1 shows performance gaps between Sort-Loss-only and Hybrid models, revealing sorting-related disadvantages. Both PIL and Sort Loss have imperfections, but they complement each other (acting as mutual regularizers).
>
> #### b. MS-ASR Training/Inference
>
> - Errors typically occur when speakers begin with very short utterances (1–2 words).
> - This increases cpWER through incorrect speaker assignments.
> - Note: Such errors affect al diarization and MS-ASR systems, not just ours.
>
> ### 4. Relation to DOM-SOT and Comparison
>
> - "Arrival time" is not the only quantity that can be used to resolve permutations. Other options include: Total speaking time (DOM-SOT), End time, Speaking rate can be used.
>
> - However, only arrival time determines the permutation at the start and does not require future input. For example, using "total speaking time" requires input from start to end, unsuitable for streaming or divide-and-conquer scenarios.
>
> - DOM-SOT work is not tackling the same problem. DOM-SOT proposes a different loss criterion for multi-talker ASR but does not investigate supervision of a speaker diarization model.

---

### Decision · Program_Chairs · 2025-05-01

**Decision:**

Accept (poster)

**Comment:**

This paper proposes Sortformer which is an encoder-based model for speaker diarization designed for integration with multi-speaker ASR. The authors propose Sort Loss to sort speakers based on their arrival time to resolve the speaker permutation problem. Sort Loss can work either by itself or jointly with the conventional permutation invariance loss (PIL). On top of that, the authors also introduces sinusoidal kernels to encode speaker identities to provide speaker supervision in the training. As a result, multi-speaker ASR can be trained analogously to the mono-speaker ASR case. Experimental results show the effectiveness of Sortformer in improved performance on speaker diarization and multi-speaker ASR.  Most of the concerns raised by the reviewers have been cleared in the rebuttal by the authors (including an error in proving the PI property of MHA under the setting used in the work).   Overall, the work is interesting and the performance is good, especially when using the hybrid loss consisting of both sort loss and PIL.